# Representation Drift Compensation: A Zero-Cost Enhancement for LLM Decomposition

## Abstract

While low-rank decomposition offers potential for LLM size reduction, its application is limited by considerable performance degradation. In this work, we identify and formalize a key overlooked issue in LLM decomposition: *representation drift*. We show that approximation errors introduced by decomposition propagate and amplify non-linearly through the deep layers of the transformer architecture, progressively distorting internal representations and degrading downstream performance. To mitigate this, we introduce a conceptually simple but principled compensation mechanism, named "Decomper", that operates by suppressing error at its source. By learning to align the output distribution of decomposed transformer blocks with their original counterparts, our method effectively counteracts representation drift, achieving notable performance recovery with zero inference overhead. Extensive experiments across OPT, LLaMA-2, LLaMA-3, and QWen exhibit remarkable improvements in language modeling, common-sense reasoning, knowledge-based reasoning, and vision-language tasks. For instance, on LLaMA-3-8B and OPT-13B at 40% compression, perplexity is reduced by more than 70% while reasoning task accuracy improves by over 10%. Our code is available at this anonymous URL.

## 1 Introduction

The scaling of large language models (LLMs) has delivered impressive capabilities but also created formidable challenges for their practical deployment due to substantial computational and memory demands (Kaplan et al., 2020; Miao et al., 2023). Consequently, model compression has become an essential research direction, with techniques like pruning (Frantar & Alistarh, 2023; Sun et al., 2024; Ashkboos et al., 2024; An et al., 2024; Ma et al., 2023), quantization (Frantar et al., 2023; Lin et al., 2024; Dettmers et al., 2023), and low-rank decomposition (Hsu et al., 2022; Yuan et al., 2023; Wang et al., 2025b;a; Yu & Wu, 2023; Chavan et al., 2024) being actively explored to reduce model size and accelerate inference.

Among these techniques, low-rank decomposition offers an approach to parameter reduction by approximating weight matrices with the product of two smaller matrices, typically achieved through Singular Value Decomposition (SVD). Despite its theoretical appeal for parameter reduction, low-rank decomposition often leads to notable performance degradation, which has hindered its practical adoption. Advanced methods primarily focus on minimizing the reconstruction error of individual weight matrices or their activations (Yuan et al., 2023; Wang et al., 2025b; Huang et al., 2025). However, as our analysis reveals, this layer-wise optimization is insufficient to prevent the accumulation and amplification of approximation errors throughout the network. We identify representation drift as the fundamental cause: minor approximation errors per layer are non-linearly amplified and accumulate across the deep network, leading to a substantial divergence in the latent representations from their original distribution. This drift manifests as a severe value reduction in the distribution of hidden states, ultimately degrading the model's quality.

In this work, we first provide a diagnosis of this problem, including a theoretical analysis of the reconstruction loss and its propagation through transformer blocks. We then introduce Decomper, a simple and effective compensation mechanism designed to mitigate this drift directly. Our key insight is that learning a compensation term for each decomposed linear layer can effectively mitigate the approximation error, dramatically reducing the initial error injected into the computational

graph. This mechanism is optimized to minimize block-level output divergence using a small calibration set (*e.g.,* 512 sequences with 256 tokens), is straightforward to implement, and adds no inference overhead as the learned compensation term can be fused into the existing linear operation.

Our comprehensive evaluations on the diverse model architectures (OPT, LLaMA-2, LLaMA-3, Qwen3 and QWen2.5-VL) across multiple scales (7B, 8B, 13B, and 30B) demonstrate that Decomper consistently outperforms strong baselines in both structured pruning and low-rank decomposition. For instance, under the 40% compression ratio with OPT-13B, Decomper outperforms SVD-LLM, yielding a perplexity reduction from 68.0 to 15.88 and downstream task accuracy enhancement from 42.6% to 52.7%. Its efficacy extends to vision-language models, yielding remarkable improvements in COCO captioning metrics, notably a 23% CIDEr gain at 20% compression.

In summary, our contributions are as follows:

- We identify and formalize the problem of representation drift as a key cause of performance degradation in low-rank decomposed LLMs.

- We propose a simple, effective, and zero-overhead compensation mechanism to mitigate this drift. To demonstrate its effectiveness, we quantitatively confirm the reduction in drift via Wasserstein distance analysis.

- We empirically validate our method's superiority over existing techniques across multiple models and benchmarks. We further demonstrate its synergistic value with quantization and its capability to recover performance on complex reasoning tasks.

## 2 RELATED WORK

In this section, we review related model compression techniques, including network pruning, quantization, and low-rank decomposition.

*Network Pruning.* Unstructured pruning focuses on exploiting the inherent sparsity of model weights to eliminate specific connections, but it often struggles to achieve significant speedup due to hardware limitations (Frantar & Alistarh, 2023; Sun et al., 2024; Dong et al., 2024; Zhang & Papyan, 2025; Makni et al., 2025). In contrast, hardware-accelerated structured pruning removes entire channels or components from LLMs (Ma et al., 2023; Ashkboos et al., 2024; van der Ouderaa et al., 2024; Lin et al., 2025; An et al., 2024). For example, SliceGPT (Ashkboos et al., 2024) uses a transformation matrix to eliminate rows and columns, requiring extra adapters for the new dimensions. Notably, FLAP (An et al., 2024) employs a bias compensation heuristic calculated as the mean contribution of pruned weights ($c \approx \mathbb{E}[W_{\text{pruned}}X]$). While effective for pruning, our analysis (Sec. 3) shows this is insufficient for low-rank decomposition because it only corrects the mean shift, ignoring the significant input-dependent variance error and non-linear distortions inherent to decomposition.

*Quantization.* Quantization compresses storage demands by mapping floating-point operations to lower-precision integer computations (Frantar et al., 2023; Lin et al., 2024; Dettmers et al., 2023). However, its adaptability and hardware generalization are often constrained by reliance on specific hardware support. This technique is orthogonal to other compression approaches and can be seamlessly integrated with them.

*Low-Rank Decomposition.* In the low-rank decomposition method, the weight matrix is approximated by the product of smaller matrices. One common approach uses Singular Value Decomposition (SVD) and its variations. Direct application of vanilla SVD methods often leads to a significant performance drop. FWSVD (Hsu et al., 2022) consider the importance of the parameters and use Fisher information to reconstruct the weight matrix parameters. Recent truncation-aware methods consider the activation reconstruction loss and connect singular values to compression loss (Yuan et al., 2023; Huang et al., 2025; Wang et al., 2025b;a). Another category of methods focuses on decomposition in the feature space (Chavan et al., 2024). Some works (Yu & Wu, 2023; Ji et al., 2024) introduce Atomic Feature Mimicking (AFM) to break down the output vector from the fully connected layer. However, low-rank decomposition methods encounter noticeable performance degradation, making these methods have not become a mainstream method.

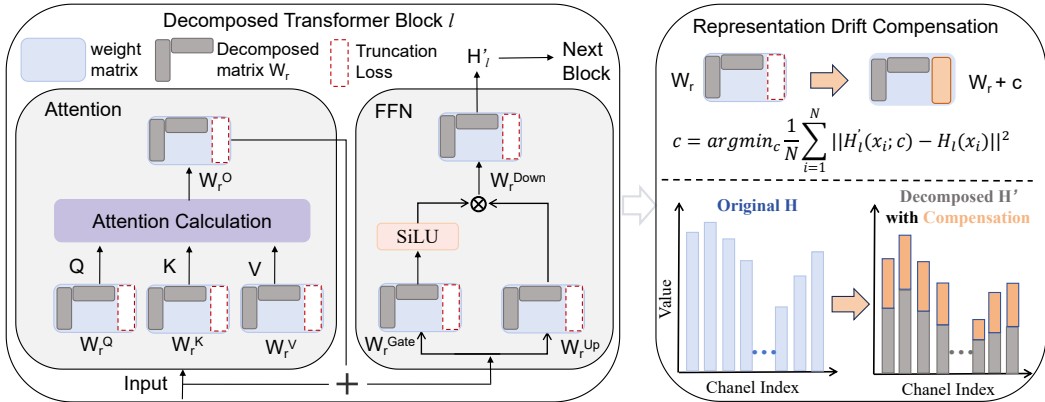

Figure 1: Decomper: A "compress-then-align" paradigm. We identify the latent representation drift that arises from low-rank decomposition techniques. Based on this finding, we propose a post-hoc compensation mechanism to mitigate the distribution drift.

## 3 REPRESENTATION DRIFT: DIAGNOSIS AND MITIGATION

We present a theoretical analysis of representation drift and quantitatively demonstrate the effect of low-rank decomposition and compensation. Our analysis proceeds in three steps: (i) quantify per-layer decomposition residuals using SVD, (ii) propagate first-order perturbations through a stack of transformer blocks using first-order Taylor expansions and operator-norm bounds, and (iii) analyze the effect of compensation and its impact on expected error.

**Notation.** We summarize the key notations used throughout our analysis for clarity. $\mathcal{F}_l$: The mapping function of the $l$-th Transformer block. $W, W_r$: Original weight matrix and its low-rank approximation. $R$: Residual error matrix ($R = W - W_r$). $H_l, H_r^l$: Outputs of the original and decomposed $l$-th block. $\mathcal{D}_{\text{drift}}^l$: Representation drift at block $l$. $\mu, \Sigma$: Mean and covariance of the input activations $X$. $c$: compensation vector for correction.

### 3.1 DIAGNOSIS: ERROR PROPAGATION AND AMPLIFICATION

**Linear Layer Reconstruction Loss** Given a weight matrix $W \in \mathbb{R}^{m \times n}$, low-rank decomposition aims to reduce parameters while minimizing the reconstruction error. We approximate $W$ by a rank-$r$ matrix $W_r = AB$, where $A \in \mathbb{R}^{m \times r}$, $B \in \mathbb{R}^{r \times n}$ are factorized via Singular Value Decomposition (SVD) as $A = U_r \sqrt{\Sigma_r}$, $B = \sqrt{\Sigma_r} V_r^\top$. Here, $U_r$ and $V_r$ contain the top-$r$ singular vectors, and $\sqrt{\Sigma_r}$ is a diagonal matrix of the square roots of the top-$r$ singular values. This reduces the parameter count from $m \times n$ to $(m + n) \times r$.

Let $W_r$ be its rank-$r$ truncated approximation and $R = W - W_r$ the residual. The reconstruction loss for input activations $X$ with mean $\mu$ and covariance $\Sigma$ is defined as:

$$\mathcal{L}(W_r) = \mathbb{E}[\|WX - W_rX\|_2^2] = \mathbb{E}[\|RX\|_2^2] = \text{Tr}\big(R(\Sigma + \mu\mu^\top)R^\top\big) = \|R\mu\|_2^2 + \text{Tr}(R\Sigma R^\top). \quad (1)$$

Please refer to Appendix A for the full derivation. This error provides a layer-wise baseline but crucially ignores how these errors interact and transform through the network's non-linearities.

**Representation Drift in the Transformer Block** A Transformer block operates as a complex dynamical system, which consists of multiple linear projections in both self-attention module $\text{SA}(X) = W^O \cdot \text{Softmax}\left(\frac{W^Q X (W^K X)^\top}{\sqrt{d_k}}\right) W^V X$ and feed-forward layers $\text{FFN}(X) = W^{\text{down}}\big(\sigma(W^{\text{up}}X) \odot W^{\text{gate}}X\big)$. After decomposition, each weight $W^j$ becomes $W^j = W_r^j + R^j$. Although existing methods minimize the reconstruction error for each individual linear layer, the output of the entire decomposed Transformer block still exhibits significant deviation from that of the original block. This divergence stems not only from the accumulation of errors across multiple layers but also from the non-linear amplification of these errors through the deep network architecture.

Using LLaMA-3-8B as an example, we investigate the distribution of latent representations for the original model, the low-rank decomposition model, and the decomposed model enhanced with our compensation mechanism. We investigate the absolute values of the latent representations from Transformer blocks, displaying the 50th percentile, 95th percentile, and maximum value for each channel. For brevity, we illustrate the outputs from the 23rd Transformer block (out of 32). As shown in the first and second sub-figures of Figure 2, despite employing advanced low-rank decomposition techniques (*e.g.,* truncation-aware data whitening minimizes linear layer reconstruction error), a significant drift of representations is observed between the decomposed and original models, particularly in the middle to tail blocks, which experience higher compression ratios. This drift results in inconsistent final outputs of the model, as the decomposed model yields latent representation values significantly lower than those of the original model. Such a pattern narrows the data distribution and negatively affects model quality. We provide additional visualizations of the latent representations for different model architectures in Appendix F.

To analyze this phenomenon, we define *representation drift* as the divergence between the hidden state distributions of the original and decomposed models across transformer blocks. Let $H^l = \mathcal{F}(H^{l-1}; \{W^i\})$ and $H_r^l = \mathcal{F}(H^{l-1}; \{W_r^i\})$ be the outputs of the $l$-th transformer block in the original and decomposed model, respectively. $\mathcal{F}$ represents the composite function of self-attention, MLP, residual connections, and LayerNorm operations. The original weights $W^i, i \in \{q, k, v, o, gate, up, down\}$ are replaced by their approximations $W_r^i$. The drift at block $l$ is quantified by the expected divergence:

$$\mathcal{D}_{\text{drift}}^l = \mathbb{E}[\|H^l - H_r^l\|_2^2]. \tag{2}$$

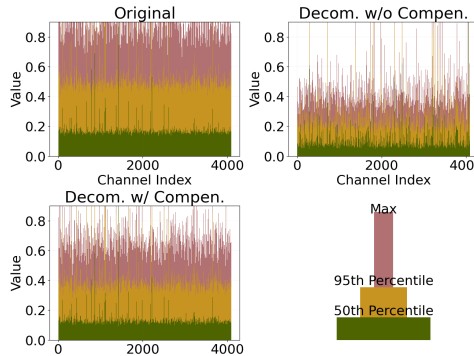

Figure 2: Latent representation drift in decomposition and our compensation on LLaMA-3-8B (23rd Block).

**Error Propagation and Amplification** To analyze the representation drift $\mathcal{D}_{\text{drift}}$, we employ a first-order Taylor expansion around the original weights $W^i$. The error propagates through the entire network. The divergence at the final block $L$ can be expressed recursively:

$$\mathcal{D}_{\text{drift}}^L = \mathbb{E}[\|H^L - H_r^L\|_2^2] \approx \mathbb{E}[\|\sum_{l=1}^{L} (\underbrace{\prod_{j=l+1}^{L} \frac{\partial H^j}{\partial H^{j-1}}}_{\text{Propagated Error}}) \cdot (\sum_{i \in \mathcal{W}} \underbrace{\frac{\partial \mathcal{F}^l}{\partial W^i} R^i}_{\text{Local Error}})\|_2^2] \tag{3}$$

where $R^i = W^i - W_r^i$ is the residual matrix for the $i$-th linear layer.

The *local error* captures the direct effect of decomposing the $i$-th linear layer in block $l$. For a linear layer $y = WX + b$, this error is $RX$, whose expected squared norm is $\mathbb{E}[\|RX\|_2^2] = \|R\mu\|_2^2 + \text{Tr}(R\Sigma R^\top)$, as derived in Equation 1. Equation 3 serves as a diagnostic tool, illustrating that the final error is a weighted sum of local errors where weights are products of Jacobians. Errors introduced in earlier layers (smaller $l$) undergo more amplification through the product of Jacobians $\prod_{j=l+1}^{L} \frac{\partial H^j}{\partial H^{j-1}}$, making them more detrimental to the final output.

## 3.2 A Simple and Effective Compensation Mechanism

**Output Alignment via Error Suppression** As derivation in Section 3.1, our key idea is to compensate for the approximation error at each linear layer directly, preventing small local errors from accumulating across layers. We aim to find a correction term such that:

$$W_r X + compensation \approx WX. \tag{4}$$

A natural choice is to use a bias vector $c$, which is efficient to learn and add. Thus, for each decomposed layer, we learn a compensation vector $c$ satisfying: $W_r X + c \approx WX$. This is equivalent to learning an approximation $c \approx (W - W_r)X = RX$. Therefore, introducing $c$ effectively reduces the local error and mitigates the error propagation:

$$\mathbb{E}[\|WX - (W_r X + c)\|_2^2] \ll \mathbb{E}[\|RX\|_2^2] \tag{5}$$

By minimizing the local error at its source, our compensation mechanism *prevents the initial introduction of substantial error* into the computational graph. This drastically reduces the error available for propagation and amplification through subsequent layers, thereby mitigating the latent representation drift and aligning $H'_l$ more closely with $H_l$.

**Proposition 1.** *Let input activations $X$ with mean $\mu$ and covariance $\Sigma$. For a decomposed linear layer with compensation vector $c$ optimized to minimize the expected reconstruction error $\mathbb{E}[\|WX - (W_r X + c)\|_2^2]$, the expected error is upper-bounded by the variance component of the residual error.*

*Proof.* Let $R = W - W_r$. The objectative is to find a constant vector $c$ that minimizes $\mathbb{E}[\|RX - c\|_2^2]$. It is a standard statistical result that the optimal solution is the expected value: $c^* = \mathbb{E}[RX] = R\mu$. Substituting $c^* = R\mu$ back into the objective function, the residual error becomes $R(X - \mu)$. The expected squared norm is derived as: $\mathbb{E}[\|R(X - \mu)\|_2^2] = \text{Tr}(R \cdot \mathbb{E}[(X - \mu)(X - \mu)^\top] \cdot R^\top) = \text{Tr}(R\Sigma R^\top)$. This confirms that the error is bounded by the variance component. $\square$

**Learned bias as a simple and practical solution**  Although Proposition 1 provides a theoretical closed-form solution $c^\star = R\mu$, its direct application in practice faces several challenges: (1) *The true input mean $\mu$ is unknown and must be estimated from a finite calibration set.* (2) *Nonlinear functions within Transformer blocks cause significant distributional shift between layers, making a single, static estimate of $\mu$ insufficient for all inputs.*

A seemingly straightforward approach would be to compute this theoretical solution empirically by estimating $\hat{\mu} = \frac{1}{N}\sum x_i$ from the calibration data and setting $\hat{c} = R\hat{\mu}$. This mirrors the heuristic used in FLAP (An et al., 2024) for pruning, where the average contribution of pruned components is used as a bias correction. However, our analysis (Eq. 1) and practice reveal this heuristic is inadequate for low-rank decomposition. As established in Proposition 1, $c^\star = R\mu$ only mitigates the mean error component $\|R\mu\|_2^2$, leaving the significant input-dependent variance term $\text{Tr}(R\Sigma R^\top)$ unaddressed. Furthermore, truncation-based low-rank decomposition introduces nonlinear operator errors that a static mean correction cannot fix. This explains why averaging tricks help pruning but are insufficient for decomposition.

Therefore, we formulate a block-level optimization problem using a small calibration dataset $x_i$ to empirically estimate the input distributions and account for non-linear interactions:

$$\hat{c} = \arg\min_c \ \frac{1}{N}\sum_{i=1}^{N} \|H_r^l(x_i; c) - H^l(x_i)\|_2^2 \qquad (6)$$

where $\{x_i\}_{i=1}^N$ are calibration samples, and $H^l$ and $H_r^l$ denote the outputs of the original and decomposed $l$-th Transformer block, respectively. As detailed in Algorithm 1 (Appendix B), we solve this efficiently by sequentially optimizing $c$ for each block via gradient descent, while keeping the original and decomposed model weights frozen.

Although the non-linearities in $\mathcal{F}$ render the optimization problem non-convex, we find its loss landscape to be well-behaved in practice, enabling the effective optimization of the compensation vector $c$. The resulting solution $\hat{c}$ provides a practical and highly effective means of minimizing the empirical output divergence. Consequently, while global optimality cannot be guaranteed, the procedure consistently converges to a strong local optimum that yields substantial performance recovery. Furthermore, this approach is highly lightweight, as it only requires optimizing low-dimensional vectors rather than full weight matrices.

Theoretically, our compensation mechanism not only reduces the immediate reconstruction error but also suppresses error propagation throughout the network. Empirically, as visually demonstrated in Figure 2, the compensated representations align more closely with those of the original model. Specifically, the compensation effectively restores the range of representation values—which narrows considerably after low-rank decomposition—to distributions much closer to those of the original model. This realignment translates into improved prediction quality and greater reliability in the decomposed model. The improvement is especially notable in middle and later blocks with high compression ratios, where representation drift is most severe in existing methods.

Quantifying Representation Alignment: To quantitatively verify the effectiveness of our drift compensation, we computed the Wasserstein distance between the latent distributions of the original and

decomposed models. As shown in Table 1, compensation significantly pulls drifting representations back to the original manifold. On LLaMA-3-8B, compensation reduced the distance from 0.1324 to 0.0614 (a 53.6% reduction), and on LLaMA-2-7B from 0.2865 to 0.1749 (39.0% reduction).

**Zero-Cost Deployment**    Following optimization, the learned compensation term, $c$, is seamlessly integrated into the linear layer by directly adding it to the existing bias term: $b' = b + c$. This design choice ensures several practical advantages: (1) the learned compensation terms are efficiently incorporated without requiring a separate parameter layer; and (2) inference speed remains unaffected, as the vector addition operation is inherently fused into the exist-

Table 1: Wasserstein distance comparison showing representation alignment recovery.

| Model | w/o | w/ | Improv. |
|---|---|---|---|
| LLaMA-3-8B | 0.1324 | 0.0614 | **53.6%** |
| LLaMA-2-7B | 0.2865 | 0.1749 | **39.0%** |
| LLaMA-2-13B | 0.1820 | 0.0946 | **48.1%** |

ing linear computation and enjoys native hardware support; (3) the computational graph preserves its original structure, ensuring compatibility with existing deployment frameworks and optimization techniques.

## 4 EXPERIMENTS

This section presents a comparative analysis against strong baselines across diverse models and benchmarks (§ 4.1). We then ablate our core contribution, demonstrate its practical utility regarding inference speedup and quantization compatibility (§ 4.2), and discuss its distinctions from other parameter-update strategies (§ 4.3). Extended results, including higher compression ratios and generality to various techniques and vision-language models, are detailed in the Appendix.

**Experimental Settings**    *Models*: We experiments with diverse models and scales, including LLaMA-2-7B/13B (Touvron et al., 2023), OPT-6.7B/13B/30B (Zhang et al., 2022), LLaMA-3-8B (Grattafiori et al., 2024), Qwen3-8B (Yang et al., 2025), and QWen2.5-VL-7B-Instruct (Bai et al., 2025). We intentionally select both older (MHA-based) and newer (GQA-based) architectures to demonstrate generalizability. *Benchmarks*: We evaluate models on: WikiText2 perplexity (Merity et al., 2017); common-sense reasoning via LMEH (BoolQ, PIQA, WinoGrande, HellaSwag, ARC, OBQA) (Gao et al., 2021); knowledge-intensive MMLU (57 challenging tasks) (Hendrycks et al., 2021); and vision-language tasks on MS COCO (Lin et al., 2014). We additionally include GSM8K (Cobbe et al., 2021) to probe complex reasoning capabilities. *Baselines*: We compare Decomper with the strong open-source structured pruning and low-rank decomposition methods, including LLM-Pruner (Ma et al., 2023), FLAP (An et al., 2024), SliceGPT (Ashkboos et al., 2024), AFM (Yu & Wu, 2023), ASVD (Yuan et al., 2023), and SVD-LLM (Wang et al., 2025b). Methods like FLAP, LLM-Pruner, and ASVD are not currently adapted for the OPT architecture, and FLAP does not support the Grouped Query Attention (GQA) architecture used in LLaMA-3.

**Implementation Details**    In the main experiments, we use data-whitening SVD as a representative instantiation. Specifically, instead of decomposing $W$ directly, we apply SVD to the whitened matrix $WS$, where $S$ captures input activation statistics (*e.g.,* covariance), thus considering the input distribution. Our method is applicable to various LLM decomposition techniques; additional results with the PCA-based method AFM are included in the Appendix due to space limitations. The compensation terms are optimized with AdamW using a cosine annealing learning rate scheduler, with an initial learning rate of 0.005. We use a calibration set of 2048 samples, each of sequence length 256. Following previous works, the compression ratio across layers is allocated proportionally to their importance measured by Fisher Information, and we do not compress the initial and terminal layers (Hsu et al., 2022; Ma et al., 2023; Huang et al., 2025). For fair comparison, our defined compression ratio is the stricter memory-based ratios rather than parameter count reduction.

### 4.1 OVERALL PERFORMANCE

**Downstream Tasks Performance**    Table 2 and 10 show that Decomper delivers consistent and remarkable performance recovery, achieved using only a general-domain Wiki corpus rather than task-related data. Our analysis reveals three key patterns: (1) *Superiority Over Strong Baselines*:

Table 2: Performance comparison of different methods on LLaMA-2-7B/13B, LLaMA-3-8B, and OPT-30B models. **Bold** and underline denote the best and second-best results, respectively. Extended results are provided in Appendix Table 10.

| Methods | Ratio | Average | BoolQ | PIQA | WinoG. | HellaS. | ARC | OBQA | MMLU |
|---|---|---|---|---|---|---|---|---|---|
| LLaMA-2-7B | 0% | 64.10 | 77.77 | 79.05 | 69.38 | 75.92 | 60.37 | 44.2 | 45.7 |
| LLM-Pruner | | 54.61 | **63.58** | **76.22** | 62.59 | **68.55** | 50.82 | **41.0** | 23.3 |
| SliceGPT | | 41.95 | 37.98 | 60.72 | 59.67 | 44.45 | 37.78 | 30.8 | 26.4 |
| FLAP | 20% | 53.18 | 53.94 | 74.54 | 62.98 | 64.74 | 48.86 | 39.6 | 31.9 |
| ASVD | | 48.88 | **63.58** | 67.57 | 61.17 | 55.80 | 41.77 | 32.8 | 26.6 |
| SVD-LLM | | 47.28 | 54.86 | 66.00 | 62.04 | 53.53 | 39.09 | 36.6 | 27.0 |
| Decomper | | **56.05** | 62.57 | 74.97 | **65.82** | 65.88 | **52.86** | **41.0** | **32.4** |
| LLaMA-2-13B | 0% | 67.55 | 80.55 | 80.41 | 72.53 | 79.41 | 63.27 | 45.6 | 55.4 |
| LLM-Pruner | | 56.39 | 62.97 | **77.97** | 60.77 | 71.26 | 55.69 | **44.0** | 22.8 |
| SliceGPT | | 44.88 | 37.86 | 62.24 | 63.54 | 47.30 | 39.25 | 38.6 | 31.0 |
| FLAP | 20% | 58.18 | 66.42 | 75.57 | 67.25 | 69.19 | 39.25 | 40.8 | 41.2 |
| ASVD | | 55.49 | **75.54** | 73.34 | 66.46 | **63.27** | 46.44 | 39.8 | 32.6 |
| SVD-LLM | | 55.65 | 70.40 | 71.76 | 68.43 | 59.59 | 49.72 | 41.0 | 34.6 |
| Decomper | | **61.90** | 71.71 | 76.82 | 69.06 | 72.66 | 59.34 | 42.2 | **44.1** |
| LLM-Pruner | | 50.28 | 62.11 | 73.18 | 57.93 | 60.89 | 44.38 | **41.4** | 22.9 |
| SliceGPT | | 39.56 | 37.83 | 56.75 | 57.70 | 38.27 | 33.53 | 31.6 | 27.1 |
| FLAP | 30% | 54.29 | **64.37** | 72.42 | 63.93 | 62.44 | 49.37 | 39.2 | 33.2 |
| ASVD | | 45.89 | 66.30 | 64.15 | 57.85 | 44.21 | 36.21 | 35.6 | 26.6 |
| SVD-LLM | | 48.27 | **64.37** | 65.61 | 62.98 | 47.81 | 39.91 | 37.6 | 28.6 |
| Decomper | | **56.39** | 63.55 | **73.72** | **66.06** | **65.71** | **54.89** | 38.2 | **34.1** |
| LLaMA-3-8B | 0% | 69.35 | 80.95 | 80.85 | 73.01 | 79.13 | 65.47 | 45.0 | 64.9 |
| LLM-Pruner | | 50.39 | 60.70 | **72.47** | 63.69 | 58.18 | 43.00 | 34.4 | 27.7 |
| SliceGPT | 20% | 37.05 | 37.83 | 54.41 | 56.75 | 35.73 | 29.06 | 28.2 | 25.4 |
| SVD-LLM | | 50.63 | **68.41** | 66.27 | 65.98 | 50.81 | 44.09 | 33.6 | 31.8 |
| Decomper | | **54.59** | 62.69 | 69.97 | **66.22** | **60.60** | **51.67** | **37.4** | **36.5** |
| OPT-30B | 0% | 57.40 | 70.12 | 78.07 | 68.59 | 72.13 | 51.71 | 39.8 | 27.1 |
| SliceGPT | | 44.00 | 38.84 | 66.59 | 58.56 | 51.97 | 37.87 | 34.4 | **25.9** |
| SVD-LLM | 30% | 53.70 | 61.10 | 76.06 | 63.69 | 68.60 | 47.86 | **40.4** | 24.6 |
| Decomper | | **55.74** | **68.50** | **76.82** | **66.46** | **69.43** | **50.29** | 38.8 | 25.3 |
| SliceGPT | | 39.20 | 37.83 | 60.77 | 51.85 | 39.02 | 33.65 | 31.4 | 25.4 |
| SVD-LLM | 40% | 51.64 | 58.26 | 73.94 | 63.46 | 64.34 | 44.82 | **38.2** | 25.3 |
| Decomper | | **54.49** | **68.38** | **75.03** | **65.11** | **66.78** | **48.88** | 37.4 | **25.5** |

Decomper outperforms not only advanced low-rank decomposition methods but also strong structured pruning baselines. For instance, on LLaMA-2-13B at 30% compression, Decomper achieves an average accuracy of 56.39, a significant improvement over the strongest baseline 50.28. (2) *General Effectiveness Across Tasks and Models*: The improvements are consistently observed across reasoning tasks, including commonsense, knowledge, and understanding. On the difficult MMLU benchmark, Decomper improves the accuracy of LLaMA-2-7B from 27.0 to 32.4. Table 4 further confirms the generality on the OPT series. (3) *Robustness Under Higher Compression*: Decomper maintains much more graceful degradation under higher compression ratios. For example, on OPT-13B at 40% compression, Decomper sustains an accuracy of 52.67, significantly reducing the performance gap to the original model compared to SVD-LLM's 42.60. We also evaluated variance over 3 random seeds for LLaMA-3-8B (20% ratio), yielding stable results: PPL $11.07 \pm 0.11$ and average accuracy $57.36 \pm 0.20$. The standard deviations are negligible compared to the performance margins we achieved over baselines. For example, the accuracy variance is only $\pm 0.2\%$, whereas our method typically outperforms baselines by $> 5\%$.

**Generation Quality**  We report the perplexity scores to evaluate generation quality. Table 3 and Table 4 show that Decomper consistently performs superior perplexity across various compression ratio settings on different model series. For instance, at a 30% compression ratio, Decomper achieves a perplexity of 6.55 for LLaMA-2-13B, outperforming FLAP's score of 7.35 and SVD-LLM's 8.45. On OPT-6.7B, Decomper reduces PPL from over 27 to 12.38. Figure 3 demonstrates that this performance improvement becomes more pronounced as the compression ratio increases, indicating a

Table 3: WikiText2 perplexity of LLaMA-2 series and LLaMA-3. The "-" means that the method is currently not applicable to the model.

| Methods | Ratio | LLaMA-2 | | LLaMA-3 |
|---|---|---|---|---|
| | | 7B | 13B | 8B |
| Dense | 0% | 5.47 | 4.88 | 6.14 |
| LLM-Pruner | | 18.88 | 21.75 | 22.47 |
| SliceGPT | | 16.51 | 13.52 | 33.67 |
| FLAP | 30% | 9.21 | 7.35 | - |
| ASVD | | 364.5 | 33.15 | - |
| SVD-LLM | | 11.52 | 8.45 | 29.22 |
| Decomper | | **8.07** | **6.55** | **16.16** |

Table 4: Perplexity and average downstream task accuracy of the OPT-6.7B/13B models.

| Methods | Ratio | PPL ($\downarrow$) | Avg. Acc. ($\uparrow$) |
|---|---|---|---|
| OPT-6.7B | 0% | 10.94 | 53.92 |
| SliceGPT | | 28.33 | 36.91 |
| SVD-LLM | 30% | 27.88 | 44.10 |
| Decomper | | **12.38** | **51.21** |
| OPT-13B | 0% | 10.13 | 54.86 |
| SliceGPT | | 18.90 | 39.83 |
| SVD-LLM | 30% | 20.75 | 45.49 |
| Decomper | | **12.94** | **52.62** |

Table 5: Comparison of perplexity and average accuracy of common-sense reasoning tasks without (w/o) and with (w/) our compensation strategy.

| Model | Dense | | Ratio | PPL ($\downarrow$) | | Avg. Acc. ($\uparrow$) | |
|---|---|---|---|---|---|---|---|
| | PPL | Avg. Acc. | | w/o | w/ | w/o | w/ |
| LLaMA-2-7B | 5.47 | 63.1 | | 9.27 | **8.07** ($\downarrow$13%) | 53.6 | **55.2** ($\uparrow$3%) |
| LLaMA-2-13B | 4.88 | 69.3 | | 7.52 | **6.55** ($\downarrow$13%) | 56.9 | **59.6** ($\uparrow$5%) |
| LLaMA-3-8B | 6.11 | 69.9 | 30% | 29.22 | **16.16** ($\downarrow$45%) | 47.3 | **50.7** ($\uparrow$7%) |
| OPT-6.7B | 10.94 | 58.0 | | 41.25 | **12.38** ($\downarrow$70%) | 47.1 | **55.1** ($\uparrow$17%) |
| OPT-13B | 10.13 | 59.1 | | 23.5 | **12.94** ($\downarrow$45%) | 48.6 | **56.6** ($\uparrow$16%) |
| OPT-30B | 9.50 | 61.7 | | 12.19 | **10.13** ($\downarrow$17%) | 57.9 | **60.1** ($\uparrow$4%) |
| LLaMA-2-7B | 5.47 | 63.1 | | 14.10 | **10.43** ($\downarrow$26%) | 45.8 | **50.0** ($\uparrow$9%) |
| LLaMA-2-13B | 4.88 | 69.3 | | 9.80 | **8.09** ($\downarrow$17%) | 51.7 | **55.8** ($\uparrow$8%) |
| LLaMA-3-8B | 6.11 | 69.9 | 40% | 89.53 | **26.66** ($\downarrow$70%) | 38.1 | **42.7** ($\uparrow$12%) |
| OPT-6.7B | 10.94 | 58.0 | | 153 | **14.25** ($\downarrow$91%) | 39.4 | **52.0** ($\uparrow$32%) |
| OPT-13B | 10.13 | 59.1 | | 68.0 | **15.88** ($\downarrow$77%) | 42.6 | **52.7** ($\uparrow$24%) |
| OPT-30B | 9.50 | 61.7 | | 13.8 | **10.75** ($\downarrow$22%) | 55.4 | **58.6** ($\uparrow$6%) |

robust capability to maintain model generation quality. Extended results are shown in Appendix Table 10.

## 4.2 IN-DEPTH ANALYSIS

This section presents a comprehensive analysis of our compensation strategy. We first quantify the contribution of the compensation mechanism itself, then examine its robustness to calibration data variations. Furthermore, we demonstrate the practical utility of our approach through measurements of inference acceleration and compatibility with quantization techniques. Finally, we validate the generality of our method across different decomposition techniques.

**Contribution of Compensation Strategy** As demonstrated in Table 5, our compensation strategy consistently improves both generation quality (measured by perplexity) and LMEH downstream task accuracy across various model architectures and compression ratios. Notable improvements include the LLaMA-3-8B model, which achieves a reduction in perplexity from 29.22 to 16.16 (a 45% relative improvement) at 30% compression ratio, and the OPT-13B model, which shows an accuracy increase

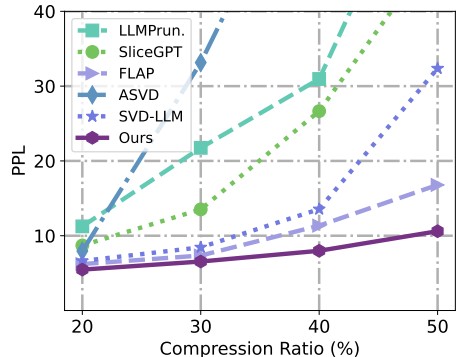

Figure 3: Perplexiton LLaMA-2-13B under 20% to 50% compression ratio.

from 48.6 to 56.6 (16% relative improvement). The benefits are even more substantial at higher

Table 6: Perplexity of LLaMA-2-13B under 30% compression ratio using calibration data with different numbers and types.

| Calibration Data | Number | | |
|---|---|---|---|
| | 512 | 1024 | 2048 |
| WikiText2 | 6.95 | 6.91 | 6.87 |
| C4 | 7.34 | 7.43 | 7.40 |

Table 7: Inference speedup of time to first token.

| Ratio | Time To First Token (speedup) | |
|---|---|---|
| | LLaMA-2-7B | LLaMA-2-13B |
| 0% | 53.34 | 88.08 |
| 30% | 46.44 (1.15×) | 70.54 (1.25×) |
| 50% | 43.42 (1.23×) | 59.26 (1.49×) |

Table 8: The synergy between decomposition and quantization is evident: Decomper (20%) + 4-bit achieves a lower perplexity than 3-bit quantization at a smaller memory footprint.

| Method | Dense | GPTQ-4bit | GPTQ-3bit | Decomper + 4bit |
|---|---|---|---|---|
| | LLaMA-2-7B | | | |
| Memory (GB) | 12.55 | 3.7 | 2.8 | 2.5 |
| PPL ($\downarrow$) | 5.47 | 6.22 | 6.96 | **6.55** |
| | LLaMA-3-8B | | | |
| Memory (GB) | 14.96 | 5.4 | 4.6 | 4.5 |
| PPL ($\downarrow$) | 6.14 | 10.49 | 16.58 | **11.72** |

compression ratios: for instance, OPT-6.7B at 40% compression ratio exhibits a dramatic perplexity reduction from 153 to 14.25 (91% improvement) while accuracy improves from 39.4 to 52.0 (32% relative improvement). These results confirm that our compensation strategy effectively mitigates performance degradation in decomposed models, with particularly significant gains observed under aggressive compression settings.

**Robustness to Calibration Dataset** We analyze the effect of the calibration data, which corrects the latent representation distribution to preserve the model quality. Table 6 illustrates the perplexity score assessed on the WikiText2 test corpus, derived from variations in quantity and category of calibration data. The variations in language modelling performance due to the calibration dataset are minor, with perplexity from 6.87 to 7.43, indicating the number and type of calibration data have a limited impact on Decomper, and a small set of calibration data (e.g., 512 samples) can bridge the latent representation gap.

**Inference Speedup** We measure the time-to-first-token (TTFT) latency on LLaMA-2 models using a sequence length of 512. As shown in Table 7, Decomper achieves notable speedups: a 1.25× acceleration at 30% compression and 1.49× at 50% compression for LLaMA-2-13B. Notably, this speedup is achieved entirely through the reduction of parameters and FLOPs resulting from decomposition. By leveraging the native hardware support for linear operations, the compensation mechanism itself contributes zero additional latency, as the $c$ vectors are fused into the bias terms and require no extra computation during inference.

**Synergy with Quantization** Quantization (e.g., FP16 to INT4) primarily targets the numerical precision, yet realizing high-performance inference with low-bit kernels typically relies on recent GPU architectures. In contrast, low-rank decomposition provides hardware-agnostic FLOPs reduction, which is orthogonal to the numerical compression of quantization. We demonstrate this synergy in Table 8. Combining 20% Decomper with 4-bit quantization achieves a superior trade-off than aggressive 3-bit quantization alone. For LLaMA-2-7B, it uses less memory (2.5 GB vs. 2.8 GB) while also achieving a better PPL (6.55 vs. 6.96). Similarly, for LLaMA-3-8B, the combined model is 41% better in PPL(11.72 vs 16.58) at comparable memory size (4.5 GB vs. 4.6 GB).

**Performance on Complex Reasoning Tasks** Complex reasoning remains a challenging frontier in structured compression. We evaluated Decomper on the GSM8K benchmark (Table 15 in the Appendix). While all compression methods underperform compared to dense baselines—reflecting a known industry-wide challenge—Decomper consistently achieves higher recovery rates than ASVD and SVD-LLM. Notably, on LLaMA-3-8B, our method attains 9% accuracy, whereas baselines drop

to nearly zero. This underscores the critical importance of drift mitigation, even in highly sensitive reasoning tasks.

**Generality to Decomposition Techniques**  It is noteworthy that Decomper is a general method. Although instantiated with SVD in our main experiments, we also provide results with a PCA-based method AFM (Yu & Wu, 2023) in the Appendix Table 11, demonstrating its consistent effectiveness across different decomposition techniques.

**Extend to Vision Language Models**  Beyond large language models and textual benchmarks, we validate our approach on vision-language tasks. For QWen2.5-VL-Instruct compressed at 20%, our compensation improves COCO captioning quality across multiple metrics, with CIDEr increasing by 23.2% and Bleu scores showing consistent gains (see Appendix Table 12), confirming the method's versatility across modalities.

### 4.3    COMPARISON WITH FINE-TUNING AND MATRIX UPDATES.

We contrast our lightweight compensation with more parameter-intensive approaches: recovery fine-tuning (FT) and least-squares (LS) updates of decomposed matrices. For the compensation mechanism, recovery fine-tuning, and decomposed matrix updates, we use 2048, 8192, and 512 samples drawn from the general-domain WikiText-2 dataset with sequence lengths of 256, 1024, and 1024, respectively. We evaluate performance using Perplexity and Average Accuracy (averaged across the 8 reasoning and knowledge benchmarks detailed in Section 4.1). As shown in Table 9, our method, despite using far less compute for adaptation, outperforms these approaches, particularly at high compression ratios. This highlights the efficiency and superiority of our targeted compensation strategy over globally modifying weights. The fact that a simple, post-hoc bias correction can outperform these methods underscores the critical importance of directly addressing representation drift, rather than attempting to fine-tune away the task error.

Table 9: Different methods for LLaMA-2 series.  The FT means recovery fine-tuning.  The LS indicates decomposed matrices updates using the least squares method.

| Method | Ratio | LLaMA-2-7B | | LLaMA-2-13B | |
|---|---|---|---|---|---|
| | | PPL ($\downarrow$) | Avg. Acc. ($\uparrow$) | PPL ($\downarrow$) | Avg. Acc. ($\uparrow$) |
| FT (SliceGPT) | | 9.39 | 41.4 | 12.9 | 46.8 |
| FT (LLM-Pruner) | 30% | 9.21 | 49.1 | 9.01 | 51.9 |
| LS (SVD-LLM) | | 11.5 | 45.4 | 8.45 | 51.2 |
| Our Compen. | | **8.07** | **51.7** | **6.55** | **56.4** |
| FT (SliceGPT) | | 13.6 | 37.9 | 14.7 | 40.9 |
| FT (LLM-Pruner) | 40% | 12.9 | 49.9 | 13.3 | 46.9 |
| LS (SVD-LLM) | | 22.3 | 39.0 | 13.2 | 45.2 |
| Our Compen. | | **10.4** | **50.0** | **7.99** | **55.8** |

## 5    CONCLUSION

In this work, we first diagnose a fundamental obstacle hindering the effectiveness of low-rank decomposition for LLMs: the phenomenon of representation drift, where approximation errors in individual layers propagate and are non-linearly amplified through the transformer architecture. To mitigate this issue, we introduced a simple and effective compensation mechanism without additional inference overhead. Our method learns compensation terms that mitigate the initial injection of error and its subsequent propagation, effectively aligning the outputs of decomposed transformer blocks with their original counterparts. Extensive experiments demonstrate that our proposed compensation strategy consistently enhances the performance of decomposed models and exhibits strong compatibility and generality.

## REPRODUCIBILITY STATEMENT

We have taken several measures to facilitate reproducibility. First, we provide experimental settings and details in the experiment section and a pseudocode implementation of our proposed Decomper method in the Appendix, which clearly outlines the key steps of our decomposition and compensation approach. Second, we have made our codebase publicly available at the anonymous URL provided in the abstract, which includes implementations of low-rank decomposition, compensation mechanisms, and evaluation scripts used in our experiments.

## LLM USAGE

The ChatGPT usage was limited to surface-level grammatical suggestions comparable to conventional spelling checkers, without involvement in substantive content generation or data analysis.

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

## A  DERIVATION OF RECONSTRUCTION LOSS

Here we provide the detailed derivation of the reconstruction loss presented in Equation 1. We begin by defining the loss function $\mathcal{L}(W_r)$ as the expected squared Euclidean norm of the error vector $RX$, where $R = W - W_r$ represents the residual matrix and $X$ is the input activations. To tractably analyze this expectation, we utilize the properties of the trace operator and the statistical moments of $X$. The derivation proceeds as follows:

$$
\begin{aligned}
\mathcal{L}(W_r) &= \mathbb{E}[\|RX\|_2^2] \\
&= \mathbb{E}[\mathrm{Tr}((RX)(RX)^\top)] \quad (\text{since } \|v\|_2^2 = \mathrm{Tr}(vv^\top)) \\
&= \mathbb{E}[\mathrm{Tr}(RXX^\top R^\top)] \\
&= \mathrm{Tr}(\mathbb{E}[RXX^\top R^\top]) \quad (\text{Linearity of Trace and Expectation}) \\
&= \mathrm{Tr}(R \cdot \mathbb{E}[XX^\top] \cdot R^\top) \\
&= \mathrm{Tr}(R(\Sigma + \mu\mu^\top)R^\top) \quad (\text{since } \mathbb{E}[XX^\top] = \Sigma + \mu\mu^\top) \\
&= \mathrm{Tr}(R\Sigma R^\top + R\mu\mu^\top R^\top) \\
&= \mathrm{Tr}(R\Sigma R^\top) + \mathrm{Tr}(R\mu\mu^\top R^\top) \\
&= \mathrm{Tr}(R\Sigma R^\top) + \mathrm{Tr}((R\mu)(R\mu)^\top) \quad (\text{Trace property}) \\
&= \mathrm{Tr}(R\Sigma R^\top) + \|R\mu\|_2^2
\end{aligned}
$$

This derivation explicitly shows that the total error consists of a mean-shift component $\|R\mu\|_2^2$ and a variance-dependent component $\mathrm{Tr}(R\Sigma R^\top)$.

## B  PSEUDOCODE OF DECOMPER

Algorithm 1 presents Decomper, a novel method for compressing pre-trained Large Language Models (LLMs) while mitigating the performance degradation typically associated with weight reduction. Decomper leverages Singular Value Decomposition (SVD) to compress the weight matrices of linear layers within each Transformer block, followed by a representation drift compensation mechanism to restore the model's representational capacity. Specifically, for each linear layer $W$ (including query, key, value, output projections, and MLP layers) within each Transformer block, Decomper first performs SVD to obtain the decomposition $W = U\Sigma V^\top$. The singular values and corresponding vectors are then truncated to a lower rank $r$, resulting in a decomposed weight matrix $W_r = U_r \Sigma_r V_r^\top$. This compression inevitably introduces a residual error $R = W - W_r$. To compensate for the representation drift induced by this compression, Decomper introduces a compensation vector $c$, which is optimized using a calibration dataset $D$. The compensation vector is fused into the bias term of the linear layer, effectively shifting the layer's output. The optimization process aims to minimize the discrepancy between the original block output $H^l(x_i)$ and the decomposed block output with compensation $H_r^l(x_i; c)$, as measured by the $L_2$ norm. The compensation vector is iteratively updated using gradient descent with learning rate $\eta$ over $E$ epochs. The loss function is defined as $\mathcal{L} = \frac{1}{N} \sum_{i=1}^{N} \|H_r^l(x_i; c) - H^l(x_i)\|_2^2$. Finally, the original weight matrix $W$ is replaced with the decomposed matrix $W_r$, and the original bias term $b$ is replaced with the compensated bias term $b' = b + c$. This process is repeated for all linear layers in all Transformer blocks, resulting in a decomposed model with compensated weights that exhibits improved performance compared to naive SVD compression.

---

**Algorithm 1** Decomper: LLM Decomposition with Representation Drift Compensation

---

**Require:** Original LLM model, Calibration dataset $D = \{x_i\}_{i=1}^N$
1: **for** each Transformer block $l$ in the model **do**
2:     **for** each linear layer $W$ in block $l$ **do**
3:         *// SVD Decomposition*
4:         Compute SVD: $W = U\Sigma V^\top$
5:         Approximate: $W_r = U_r \Sigma_r V_r^\top$
6:         *// Representation Drift Compensation*
7:         Initialize compensation vector: $c \leftarrow \mathbf{0}$
8:         **for** each batch of samples $\{x_i\}$ in $D$ **do**
9:             Compute original block output: $H^l(x_i)$
10:           Compute decomposed block output with compensation: $H_r^l(x_i; c)$
11:           Calculate loss: $\mathcal{L} = \frac{1}{N}\sum_{i=1}^N \|H_r^l(x_i; c) - H^l(x_i)\|_2^2$
12:           Update compensation vector: $c \leftarrow c - \eta\nabla_c \mathcal{L}$
13:         **end for**
14:         *// Apply Compensation*
15:         Fuse compensation into bias: $b' = b + c$
16:     **end for**
17: **end for**
18: **return** Decomposed model with compensated weights

---

## C   EXTENDED RESULTS ON HIGHER COMPRESSION RATIOS

Table 10 provides comprehensive results across multiple compression ratios (20%, 30%, and 40%) for various model families, including Qwen3-8B, OPT-6.7B, OPT-13B, LLaMA-3-8B, LLaMA-2-7B, and LLaMA-2-13B. The LLM-Pruner, FLAP, and ASVD are not applicable to the OPT architecture. These extended results demonstrate the robustness of Decomper under different compression settings. In particular, Decomper consistently achieves the best or competitive performance in both perplexity (PPL) and downstream task accuracy, even at higher compression ratios (e.g., 40%). For instance, on OPT-6.7B at 40% compression, Decomper maintains strong performance with a PPL of 14.25 and an average accuracy of 51.97. This significantly surpasses the performance of SVD-LLM, which yields a PPL of 153 and an accuracy of only 39.40 under the same conditions. Similarly, on LLaMA-2-7B at 40% compression, Decomper attains a perplexity of 10.43 and an average accuracy of 50.00, significantly outperforming methods like ASVD (PPL NaN, Acc 37.94) and SVD-LLM (PPL 14.10, Acc 45.87). These results further validate the effectiveness and stability of the proposed compensation mechanism in preserving model capabilities under different levels of compression.

## D   GENERALIZATION TO PCA-BASED DECOMPOSITION

To further validate the generality of our compensation framework, we integrate it with AFM, a PCA-based decomposition method (Yu & Wu, 2023). As summarized in Table 11, our compensation brings consistent and substantial improvements to AFM across models and compression ratios. On LLaMA-2-7B at 20% compression, perplexity drops from 8.41 to 6.70 and average accuracy rises from 55.38 to 59.86. More notably, under higher compression (40%), compensation recovers perplexity from 18.82 to 10.44 and accuracy from 42.05 to 50.16—narrowing the gap to the original model significantly. Similar trends hold for LLaMA-2-13B, where at 30% compression, accuracy improves from 49.53 to 58.93. These results confirm that our approach is decomposition-agnostic and effectively mitigates representation drift across diverse low-rank compression techniques.

## E   VISION LANGUAGE MODEL APPLICATION

To evaluate the generalizability of our method beyond pure language modeling, we additionally evaluate on vision-language tasks. Specifically, we compress the QWen2.5-VL-Instruct model and report its performance on the COCO dataset, a comprehensive benchmark for image understanding that involves object detection, segmentation, and captioning. This evaluation serves to validate

Table 10: WikiText2 perplexity and reasoning task accuracy of the decomposed models at different compression ratios.

| Methods | Ratio | PPL (↓) | Avg. (↑) | BoolQ | PIQA | WinoG. | HellaS. | ARC | OBQA |
|---|---|---|---|---|---|---|---|---|---|
| Qwen3-8B | 0% | 9.25 | 69.51 | 86.61 | 77.69 | 67.96 | 74.84 | 40.73 | 42.0 |
| SVD-LLM | 20% | 13.86 | 56.91 | **79.08** | 69.42 | 61.09 | 54.93 | 48.71 | **36.4** |
| Decomper | | **13.33** | **58.99** | 77.89 | **69.59** | **64.01** | **57.88** | **54.28** | 35.0 |
| SVD-LLM | 30% | 20.03 | 46.45 | 63.12 | 62.30 | 54.62 | 41.46 | 36.94 | 29.8 |
| Decomper | | **14.85** | **55.66** | **75.05** | **67.14** | **61.96** | **52.98** | **49.16** | **34.2** |
| SVD-LLM | 40% | 42.82 | 38.25 | 37.86 | 56.75 | 52.33 | 34.13 | 29.66 | 27.4 |
| Decomper | | **19.06** | **48.51** | **62.91** | **61.37** | **58.88** | **45.06** | **40.78** | **29.8** |
| OPT-6.7B | 0% | 10.94 | 58.04 | 70.12 | 78.07 | 68.59 | 72.13 | 51.71 | 39.8 |
| SliceGPT | | 17.77 | 43.41 | 37.86 | 61.59 | 57.22 | 41.63 | 35.99 | 33.6 |
| SVD-LLM | 20% | 16.38 | 51.55 | 57.52 | 71.16 | 59.43 | 54.26 | 41.14 | 36.2 |
| Decomper | | **11.44** | **55.97** | **60.83** | **75.41** | **62.90** | **64.79** | **45.84** | **36.2** |
| SliceGPT | | 28.83 | 38.65 | 37.83 | 55.01 | 53.43 | 33.25 | 31.51 | 28.0 |
| SVD-LLM | 30% | 27.88 | 47.13 | 57.19 | 67.57 | 55.56 | 41.35 | 38.12 | 32.0 |
| Decomper | | **12.38** | **55.06** | **65.35** | **73.23** | **62.19** | **60.82** | **44.42** | **35.0** |
| SliceGPT | | 65.38 | 35.07 | 37.83 | 50.82 | 49.25 | 28.85 | 26.06 | 26.6 |
| SVD-LLM | 40% | 153 | 39.40 | 40.18 | 58.54 | 50.51 | 31.58 | 32.78 | 29.4 |
| Decomper | | **14.25** | **51.97** | **60.86** | **70.78** | **59.98** | **55.88** | **40.64** | **35.0** |
| OPT-13B | 0% | 10.13 | 59.12 | 65.29 | 76.82 | 64.88 | 69.79 | 48.94 | 39.2 |
| SlicGPT | | 13.78 | 46.50 | 39.54 | 66.59 | 58.80 | 51.03 | 37.97 | 33.6 |
| SVD-LLM | 20% | 12.75 | 54.58 | 52.17 | 75.19 | 62.19 | 65.68 | 45.03 | 36.8 |
| Decomper | | **10.25** | **56.94** | **59.20** | **75.79** | **64.09** | **68.69** | **47.12** | 36.6 |
| SlicGPT | | 18.90 | 41.93 | 37.95 | 61.37 | 53.99 | 40.20 | 34.61 | 30.8 |
| SVD-LLM | 30% | 20.75 | 48.58 | 55.17 | 65.67 | 53.75 | 51.89 | 39.98 | 33.6 |
| Decomper | | **12.94** | **56.62** | **64.74** | **73.39** | **62.43** | **65.18** | **45.52** | **39.6** |
| SlicGPT | | 39.09 | 37.14 | 37.83 | 54.03 | 51.85 | 29.95 | 28.96 | 28.4 |
| SVD-LLM | 40% | 68.00 | 42.60 | 54.04 | 61.26 | 49.49 | 38.77 | 33.03 | 28.6 |
| Decomper | | **15.88** | **52.67** | **58.69** | **70.73** | **59.04** | **59.98** | **42.64** | **35.0** |
| LLaMA-3-8B | 0% | 6.14 | 69.98 | 80.95 | 80.85 | 73.01 | 79.13 | 65.47 | 45.0 |
| LLM-Pruner | | 22.47 | 46.73 | 53.30 | **67.90** | 57.85 | 47.92 | 33.77 | 32.6 |
| SliceGPT | 30% | 33.67 | 35.70 | 37.83 | 51.90 | 51.14 | 30.46 | 25.99 | 26.6 |
| SVD-LLM | | 29.22 | 47.33 | 61.28 | 61.43 | 59.19 | 41.71 | 39.35 | 29.0 |
| Decomper | | **17.26** | **50.65** | **63.49** | 65.07 | **61.33** | **48.63** | **41.60** | **32.8** |
| LLM-Pruner | | 51.58 | 42.15 | **52.39** | 62.08 | 55.09 | 38.67 | 28.10 | **30.6** |
| SliceGPT | 40% | 60.20 | 35.05 | 37.83 | 51.85 | 50.51 | 28.50 | 25.02 | 26.6 |
| SVD-LLM | | 89.53 | 38.09 | 41.07 | 55.93 | 53.75 | 30.88 | 29.51 | 26.0 |
| Decomper | | **26.66** | **42.65** | 47.13 | 59.68 | **58.41** | 38.72 | 33.02 | 28.6 |
| LLaMA-2-7B | 0% | 5.47 | 66.72 | 77.77 | 79.05 | 69.38 | 75.92 | 60.37 | 44.2 |
| LLM-Pruner | | 18.88 | 50.96 | 52.84 | **72.00** | 54.62 | 56.94 | 41.44 | 37.0 |
| SliceGPT | | 16.51 | 39.03 | 37.83 | 55.60 | 54.54 | 35.06 | 31.67 | 28.4 |
| FLAP | 30% | 8.91 | 52.11 | 52.20 | 70.29 | 60.06 | 56.58 | 43.72 | 38.2 |
| ASVD | | 33.15 | 46.41 | 50.70 | 52.56 | 52.01 | 28.76 | 26.24 | 22.8 |
| SVD-LLM | | 11.52 | 45.37 | 49.88 | 61.04 | 58.64 | 43.48 | 34.98 | 34.6 |
| Decomper | | **8.07** | **55.16** | **62.35** | 70.29 | **62.98** | **58.47** | **46.82** | **38.4** |
| LLM-Pruner | | 61.16 | 44.90 | 59.85 | 64.04 | 53.51 | 40.29 | 33.09 | 31.4 |
| SliceGPT | | 27.41 | 36.31 | 37.83 | 52.23 | 51.46 | 30.92 | 27.99 | 25.8 |
| FLAP | 40% | 14.27 | 44.77 | 41.71 | **66.10** | 54.22 | **47.50** | 34.24 | **35.4** |
| ASVD | | Nan | 37.94 | 37.98 | 49.40 | 51.30 | 26.27 | 26.03 | 24.4 |
| SVD-LLM | | 14.10 | 45.87 | 51.87 | 62.19 | 59.04 | 42.86 | 36.16 | 32.8 |
| Decomper | | **10.43** | **50.00** | **61.99** | 65.13 | **59.12** | 49.25 | **39.95** | 34.6 |
| LLaMA-2-13B | 0% | 4.88 | 69.30 | 80.55 | 80.41 | 72.53 | 79.41 | 63.27 | 45.6 |
| LLM-Pruner | | 30.95 | 39.91 | 39.66 | 61.92 | 52.17 | 38.66 | 27.89 | 31.2 |
| SliceGPT | | 26.66 | 36.94 | 37.83 | 52.77 | 52.88 | 30.74 | 28.38 | 27.6 |
| FLAP | 40% | 11.35 | 54.14 | 63.39 | 69.21 | 60.93 | 55.99 | 45.12 | **39.2** |
| ASVD | | 462.4 | 37.94 | 57.71 | 54.08 | 49.64 | 28.26 | 26.24 | 23.4 |
| SVD-LLM | | 13.55 | 51.71 | 61.74 | 64.96 | 62.12 | 50.07 | 43.05 | 37.0 |
| Decomper | | **7.99** | **55.83** | 62.51 | 68.34 | **62.43** | **58.23** | **50.57** | 38.2 |

Table 11: Performance of PCA-based decomposition (AFM) with and without compensation on LLaMA-2 models across compression ratios.

| Methods | Ratio | PPL ($\downarrow$) | Avg. ($\uparrow$) | BoolQ | PIQA | WinoG. | HellaS. | ARC | OBQA |
|---|---|---|---|---|---|---|---|---|---|
| LLaMA-2-7B | 0% | 5.47 | 66.72 | 77.77 | 79.05 | 69.38 | 75.92 | 60.37 | 44.2 |
| AFM | 20% | 8.41 | 55.38 | **65.75** | 68.72 | 64.88 | 54.34 | 47.68 | 38.6 |
| w/ Compen. | | **6.70** | **59.86** | 62.57 | **74.81** | **65.67** | **66.94** | **53.82** | **41.4** |
| AFM | 30% | 11.43 | 49.53 | **64.01** | 62.13 | 61.80 | 45.16 | 40.51 | 32.6 |
| w/ Compen. | | **7.96** | **55.53** | 62.26 | **70.18** | **62.43** | **58.52** | **48.07** | **39.2** |
| AFM | 40% | 18.82 | 42.05 | 49.69 | 56.58 | 57.30 | 36.38 | 32.81 | 28.8 |
| w/ Compen. | | **10.44** | **50.16** | **61.1** | **65.13** | **59.43** | **49.0** | **41.13** | **34.2** |
| LLaMA-2-13B | 0% | 4.88 | 69.30 | 80.55 | 80.41 | 72.53 | 79.41 | 63.27 | 45.6 |
| AFM | 20% | 8.41 | 59.09 | **64.10** | 73.18 | **67.80** | 60.75 | 54.41 | 39.0 |
| w/ Compen. | | **5.97** | **62.66** | 64.04 | **76.82** | 67.72 | **71.44** | **58.50** | **41.6** |
| AFM | 30% | 11.43 | 49.53 | **64.01** | 62.13 | 61.80 | 45.16 | 40.51 | 32.6 |
| w/ Compen. | | **7.08** | **58.93** | 62.29 | **73.01** | **66.30** | **64.57** | **53.97** | **38.4** |
| AFM | 40% | 13.22 | 46.34 | 55.54 | 59.25 | 60.06 | 41.24 | 37.35 | 33.6 |
| w/ Compen. | | **8.61** | **55.20** | **61.99** | **68.34** | **64.25** | **57.10** | **48.68** | **37.4** |

Table 12: COCO evaluation results of QWen2.5-VL-Instruct at 20% compression ratio.

| Metric | w/o Compen. | w/ Compen. |
|---|---|---|
| Bleu_1 | 0.4342 | **0.4481** ($\uparrow$3.2%) |
| Bleu_2 | 0.2802 | **0.2986** ($\uparrow$6.6%) |
| Bleu_3 | 0.1776 | **0.1926** ($\uparrow$8.4%) |
| Bleu_4 | 0.1126 | **0.1244** ($\uparrow$10.5%) |
| CIDEr | 0.2973 | **0.3662** ($\uparrow$23.2%) |

whether our compensation mechanism effectively preserves capabilities across fundamentally distinct modalities and task structures.

The COCO evaluation results of the compressed QWen2.5-VL-Instruct model (20% ratio), presented in Table 12, show that our compensation brings substantial gains in captioning quality, with CIDEr increasing from 0.2973 to 0.3662 (a 23% relative improvement) and Bleu-4 from 0.1126 to 0.1244. Bleu-n (Bilingual Evaluation Understudy) measures n-gram precision between generated and reference captions, with higher scores indicating greater similarity. CIDEr (Consensus-based Image Description Evaluation), specifically designed for image captioning, assesses the consensus between generated and reference descriptions using TF-IDF weighted n-grams; higher CIDEr scores reflect more relevant and informative captions. This consistent pattern of recovery—observed in common-sense reasoning, knowledge-intensive question answering, and complex vision-language tasks—collectively supports that our approach effectively mitigates representation drift, thereby preserving semantic fidelity in low-rank LLM decomposition.

## F   ADDITIONAL LATENT REPRESENTATION DISTRIBUTIONS

In this appendix, we extend our investigation of representation drift to the LLaMA-2-7B and LLaMA-2-13B models to further illustrate the effects of low-rank decomposition and the benefits of our compensation mechanism. To highlight the effect of error propagation and amplification—which becomes more severe in deeper layers due to the cumulative effect of approximation errors through the network—we deliberately analyze Transformer blocks located in the middle-to-late stages of each model. Specifically, we visualize the absolute values of the latent representations (showing the 50th percentile, 95th percentile, and maximum value per channel) for: LLaMA-3-8B: The 23rd Transformer block. LLaMA-2-7B: The 25th Transformer block (out of 32 total blocks). LLaMA-2-13B: The 33rd Transformer block (out of 40 total blocks).

These figures mirror the analysis presented in Figure 4 for LLaMA-3-8B and demonstrate similar trends across different model architectures. The distribution of latent representations for the original models, the low-rank decomposed models, and the models enhanced with our compensation

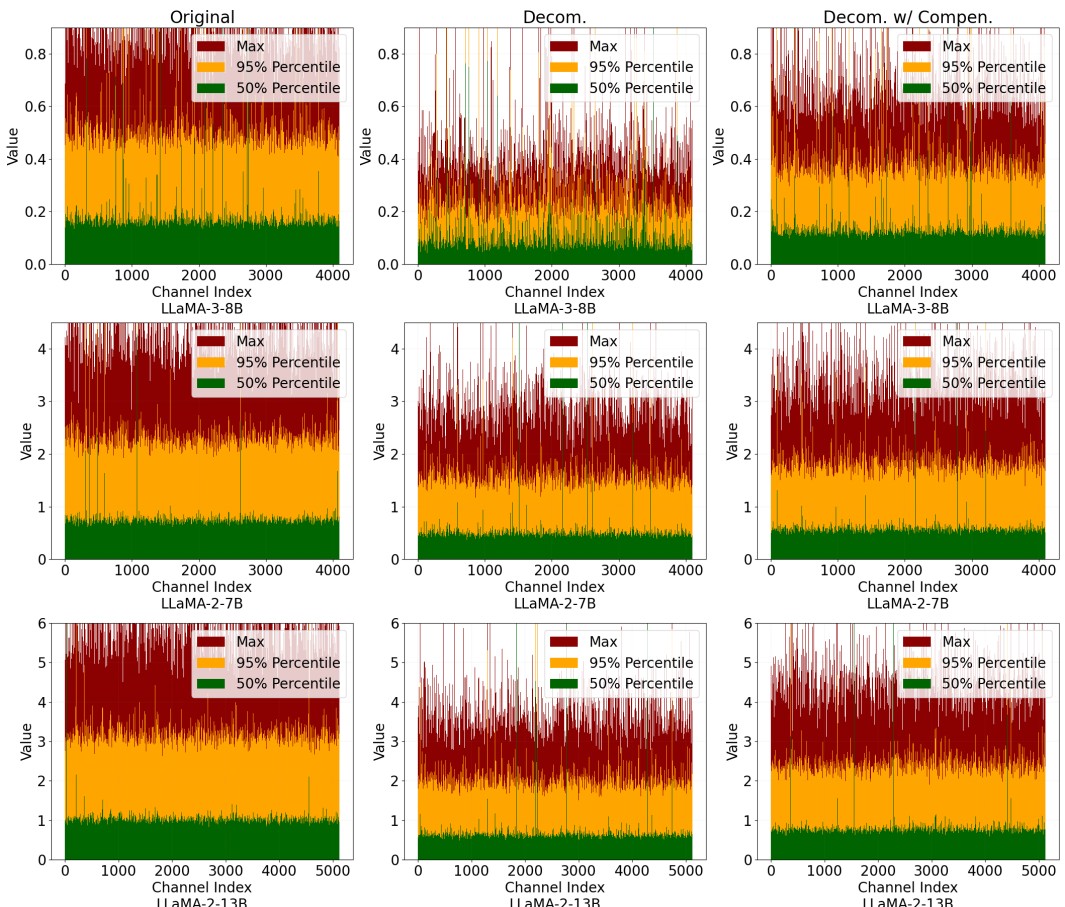

Figure 4: Latent representation drift in LLaMA-3-8B, LLaMA-2-7B, and LLaMA-2-13B decomposition and our compensation. Columns 1, 2, and 3 show the latent representations (Transformer Block outputs) of the original model, the decomposed model, and the decomposed model with compensation, respectively.

mechanism are shown. The figures display the 50th percentile, 95th percentile, and maximum value for each channel across the Transformer blocks. Our compensation mechanism effectively counteracts this drift, restoring the distribution of latent values closer to that of the original model. This recovery is evident across all percentiles, supporting our claim that the proposed method mitigates representation degradation and contributes to improved model performance. The consistent pattern of representation drift and its mitigation through our compensation strategy across various models further validates the general applicability of our approach.

## G   OPTIMIZATION EFFICIENCY AND CONVERGENCE

While the optimization problem in Equation 6 is non-convex due to the Transformer architecture, we empirically observe stable and rapid convergence. For LLaMA-3-8B optimization, the block-level $L_2$ loss typically drops steeply within the first 4 batches (e.g., from 12.40 to 7.36) and quickly converges to a stable minimum (approximately 2.99, a 76% reduction). Figure 5 visualizes this stable trajectory across different architectures.

**Computational Cost.**   We further summarize the wall-clock time of the compensation optimization phase and memory overhead of the compensation vectors in Table 13.

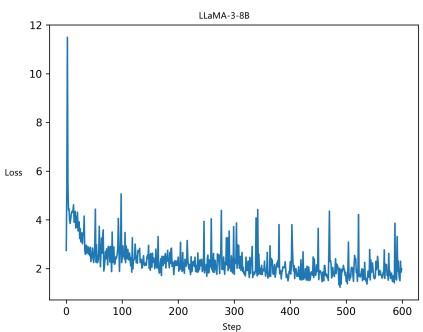

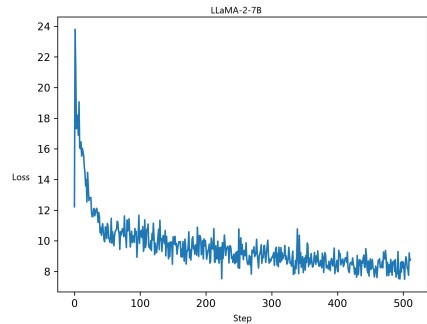

(a) Convergence curve for LLaMA-3-8B.    (b) Convergence curve for LLaMA-2-7B.

Figure 5: Empirical convergence of our optimization method across different model architectures.

- **Time Cost:** The optimization process is highly efficient as it is performed offline. Since we only optimize low-dimensional vectors $c$ (e.g., $\mathbb{R}^{4096}$) while keeping the large model weights frozen, the computational load is minimal. For instance, the process completes for LLaMA-2-13B in approximately 2 hours, with the time cost scaling roughly linearly with model size.

- **Memory Cost:** For architectures already equipped with bias terms (e.g., OPT series), our method incurs **zero** additional memory overhead as the compensation is absorbed into existing parameters. For bias-free architectures (e.g., LLaMA), the learned compensation vectors consume negligible memory (e.g., 6.5 MB for a 13B model), representing less than 0.03% of the original model size.

Table 13: Efficiency analysis showing minimal time and negligible memory overhead.

| Model | Wall-clock Time | Compensation Vectors Memory |
|---|---|---|
| OPT-6.7B | $\sim 50$ mins | 0 (Fused into existing bias) |
| OPT-13B | $\sim 80$ mins | 0 (Fused into existing bias) |
| OPT-30B | $\sim 3$ h | 0 (Fused into existing bias) |
| LLaMA-2-7B | $\sim 75$ mins | 4.4 MB |
| LLaMA-2-13B | $\sim 2$ h | 6.5 MB |

## H Ablation Study on Alignment Loss Functions

In our main methodology, we employ the $L_2$ norm as the optimization objective for the compensation mechanism. Here, we provide a justification for this choice by comparing it with other potential alignment strategies:

- **KL/JS Divergence:** These probabilistic metrics are generally unsuitable for aligning hidden states in Transformer blocks. Hidden representations often contain negative values (e.g., outputs from GeLU/SiLU activations or linear projections), which are outside the domain of logarithmic operations required by KL/JS divergence, leading to numerical instability (NaN values).

- **Wasserstein Distance:** While theoretically sound for measuring distribution shifts, estimating the Wasserstein distance is computationally expensive and difficult for efficient optimization. In contrast, $L_2$ loss offers the ideal balance.

- **Cosine Similarity:** We empirically compared our $L_2$ loss with Cosine Similarity loss ($\mathcal{L} = 1 - \cos(H, H_r)$). As shown in Table 14, while Cosine Similarity is a reasonable alternative, the $L_2$ loss consistently yields better perplexity and downstream accuracy.

Table 14: Comparison of different loss functions for the compensation mechanism on LLaMA-2-7B.

| Model | Loss Function | 20% Compression | | 30% Compression | |
|---|---|---|---|---|---|
| | | PPL ($\downarrow$) | Avg. Acc. ($\uparrow$) | PPL ($\downarrow$) | Avg. Acc. ($\uparrow$) |
| LLaMA-2-7B | Cosine Similarity | 6.94 | 59.09 | 8.32 | 55.09 |
| | $L_2$ **Loss** | **6.76** | **59.43** | **8.07** | **55.16** |

## I    PERFORMANCE ON COMPLEX REASONING TASKS

Complex reasoning remains a challenging frontier for structured compression. We evaluated Decomper on GSM8K (Table 15). While all compression methods (20% compression ratio) exhibit significant drops compared to dense baselines—a known industry-wide challenge—Decomper consistently achieves superior recovery compared to ASVD and SVD-LLM. Notably, on LLaMA-3-8B, our method reaches 9% accuracy where baselines collapse to near zero. This highlights that drift mitigation is crucial even in highly sensitive tasks.

Table 15: Accuracy on GSM8K. Decomper shows superior recovery in this brittle task.

| Model | ASVD | SVD-LLM | **Decomper** |
|---|---|---|---|
| LLaMA-2-7B | 0.00 | 0.02 | **0.02** |
| LLaMA-3-8B | - | 0.03 | **0.09** |
| OPT-6.7B | - | 0.03 | **0.04** |

## J    LIMITATIONS AND FUTURE WORK

While the proposed method demonstrates superior suitability for LLM deployment in resource-constrained environments, we explicitly distinguish its intended application scenarios based on the trade-off between structural speedup and reasoning precision. **Recommended Scenarios:** Decomper is highly effective for latency-sensitive applications requiring general language understanding, knowledge retrieval, and open-ended generation. In these areas, it provides essential FLOPs reduction with minimal quality loss. **Cautionary Scenarios:** For brittle, high-precision tasks like complex mathematical reasoning (e.g., GSM8K), capabilities tend to degrade rapidly under any aggressive compression—whether via decomposition or low-bit quantization. Consequently, we advise that decomposition in these domains should be applied with caution or paired with subsequent recovery instruction tuning.

The concept of representation drift and its mitigation via output alignment, while explored here in the context of low-rank decomposition, likely has broader implications for model compression in general. We believe it is a universal phenomenon in model compression. Our work offers a potential starting point for further research into "compress-then-align" strategies. Future work could explore automated compensation strategies for various compression techniques, including low-rank decomposition and pruning, or investigate extending the alignment objective to the distributional level. We hope this work inspires further investigation into these promising directions.

