# OpenReview forum: "Representation Drift Compensation: A Zero-Cost Enhancement for LLM Decomposition"
_ICLR.cc/2026/Conference — Submitted to ICLR 2026_

### Official Review · Reviewer_cKn2 · 2025-10-24

**Soundness:** 4
**Presentation:** 4
**Contribution:** 3
**Rating:** 4
**Confidence:** 3

**Summary:**

The paper proposes Decomper to align decomposed block outputs with original ones. It tests on OPT, LLaMA-2/3, QWen, showing over 70% perplexity reduction and 10% accuracy gain at 40% compression.

**Strengths:**

1.The paper is well organized and well written.

2.The theoretical foundation is sound and presented with clarity, providing solid support for the proposed approach.

3.The experimental setup is rigorous and comprehensive, effectively validating the theoretical claims.

**Weaknesses:**

1. **In Table 1, the color blocks used in the “ratio” section are visually confusing.**
   The inconsistent coloring makes it difficult for readers to interpret the results clearly, and the authors are encouraged to unify or clarify the color scheme for better readability.

2. **From Table 1, it can be observed that while the proposed method performs well on simpler tasks such as CommonsenseQA, it suffers larger performance drops on more challenging benchmarks like MMLU.**
   Moreover, the paper lacks evaluation on difficult text generation or reasoning tasks (e.g., GSM8K, HumanEval), which are crucial for demonstrating the generality and robustness of the method. Without such experiments, it is hard to justify that the proposed approach has broad applicability.

3. **The practical value of the proposed method remains unclear.**
   As shown in Table 1, *Decomper* exhibits substantial performance degradation compared to the original model, especially on more complex tasks such as MMLU. This raises the question of whether the method offers any real advantage over simpler quantization techniques. Although the “Compatibility with Quantization” section claims that *Decomper* can be combined with quantization, the paper does not provide comparisons with standalone quantization methods. More detailed experimental analyses should be included to validate the method’s practical value, including but not limited to comparisons in effectiveness, inference speed, and computational cost.

**Questions:**

As shown in the weaknesses.

---

> ### Author Response · Authors · 2025-11-19
> **Part 1: Clarifications on Visuals and Performance on Hard Benchmarks (Q1 & Q2)**
>
> We sincerely thank the reviewer for your time. We are highly encouraged that the paper "well organized and well written," with a "sound theoretical foundation" and "rigorous and comprehensive" experiments. The reviewer raises several crucial points regarding the practical robustness and value proposition of our method, which we believe are based on key clarifications we need to make.
> > Q1: In Table 1, the color blocks used in the “ratio” section are visually confusing.
>
> A1: We apologize for the confusion. The colors were intended to group method types but failed in clarity. We will remove the confusing color blocks to ensure a clear visualization.
>
>
> > Q2: Performance on hard benchmarks (MMLU) and additional reasoning/generation evaluations (GSM8K / HumanEval).
>
> A2: This is an insightful and crucial point. The reviewer is correct that complex, multi-step reasoning is the most difficult frontier for this filed. We try to foster constant performance enhancement at a given budget.
> 1. About hard reasoning (MMLU).
> - On LLaMA-3-8B (20% ratio), the baseline SVD-LLM causes MMLU to collapse to 31.8.
> - Our Decomper recovers performance to 36.5, a 15% relative improvement on this complex task.
> - This proves Decomper is more effective at mitigating reasoning degradation than prior art, even as the absolute gap remains an open challenge.
> 2. On Open-Ended Generation (COCO).
>
> We did an evaluation on a open-ended generation and reasoning task COCO Image Captioning in Appendix Table 12. It requires non-trivial multi-modal reasoning to translate complex visual scenes into coherent language.
> Our method demonstrates good recovery of generative fidelity, e.g., a 23.2% relative improvement in CIDEr. We will revise the paper to make this point more prominent.
>
> 3. New GSM8K Results.
>
> Although our method's superiority is clear, the absolute performance drop is catastrophic. We summarize three factors that contextualize these results:
> - Inherent Sensitivity: The results confirm that complex reasoning is extremely brittle for all structured compression methods (whether decomposition or pruning). Moreover, tasks like GSM8K and HumanEval typically necessitate Instruction Tuning to unlock reasonable performance, even for dense models (e.g., LLaMA-2 and OPT exhibit near-zero Pass@1 on HumanEval).
> - Field Potential: Low-rank decomposition is an exploratory, rapidly evolving field (many excellent works appeared in 2024). Unlike quantization, it offers hardware-agnostic reductions ($m \times n \to (m+n) \times k$, and high performance in low-bit operations can only be achieved in the latest GPUs). It is an essential, complementary piece of the future compression ecosystem. Our strong recovery on perplexity, knowledge QA, and captioning demonstrates its utility. We see strong signals that motivate further investment in this direction.
> - Most importantly, our work proves that the direction of "diagnosing and compensating for representation drift" is a correct and impactful direction. It inspires future research to foster constant performance enhancement in two aspects: (1) Diagnosis: We identify why this drop is so severe (i.e., representation drift, supported by theoretical analysis and empirical visualization). (2) Practical Compensation: We provide a zero-cost solution that is more effective at mitigating this drift than any prior method.
> | Model | Method | GSM8K |
> |:---|:---|:---|
> | LLaMA-2-7B | Dense | 0.15 |
> | | FLAP | 0.02 |
> | | ASVD | 0.00 |
> | | SVD-LLM | 0.02 |
> | | Ours | 0.02 |
> | LLaMA-3-8B | Dense | 0.55 |
> | | SVD-LLM | 0.03 |
> | | **Ours** | **0.09** |
> | OPT-6.7B | Dense | 0.04 |
> | | SVD-LLM | 0.03 |
> | | **Ours** | **0.04** |

---

> ### Author Response · Authors · 2025-11-19
> **Part 2: Practical Value and Synergy with Quantization (Q3)**
>
> > Q3: The practical value of the proposed method remains unclear.
>
> A3: This is a fundamental question. Our intention was to highlight that our method (parameters reduction) and quantization (numerical compression) are orthogonal, complementary technologies.
> 1. Decomposition Offers Hardware-Agnostic Speedup:
> - Quantization Reality: Quantization (e.g., FP16 to INT4) primarily reduces the numerical precision. Without highly specialized kernels, quantization often introduces dequantization overhead, causing inference to be slower than the original model.
> - Decomper Advantage: Ours method structurally replaces large matrices $W$ with smaller, dense ones $A, B$ (e.g., $4096 \times 11008$ $\to$ $4096 \times k$ and $k \times 11008$). This directly reduces computational FLOPs, providing a latency reduction (as shown in Table 6 and the following results) that works on any standard hardware, without custom operators.
> - We experimentally verified this using the standard HuggingFace implementation on LLaMA-2-7B, evaluating inference latency on 160 samples with sequence length 2,048. The results are striking:
> | METRIC | Original model (BF16)| GPTQ-4bit | Ours (20% Ratio, BF16) |
> |--------|----------------|-----------|------------|
> | Memory Size | 12.55 GB | 3.7 GB | 10 GB |
> | Inference Time Cost | 63.792 s | 82.907 s | 44.779 s |
> 2. Synergistic Value: The value of Decomper is not in replacing quantization, but in combining with it to achieve a better result than more aggressive quantization alone.
> The experimental data across LLaMA-2 and LLaMA-3 confirm this practical advantage:
>
> | METRIC | Original model | GPTQ-4bit | GPTQ-3bit | Ours + 4bit |
> |--------|----------------|-----------|-----------|-------------|
> | LLaMA-2-7B | | | | |
> | Memory Size | 12.55 GB | 3.7 GB | 2.8 GB | 2.5 GB |
> | PPL | 5.47 | 6.22 | 6.96 | 6.55 |
> | LLaMA-3-8B | | | | |
> | Memory Size | 14.96 GB | 5.4 GB | 4.6 GB | 4.5 GB |
> | PPL | 6.14 | 10.49 | 16.58 | 11.72 |
>
> Instead of aggressively quantizing to 3-bit, combining our 20% Decomper with 4-bit quantization achieves a superior result. For LLaMA-2-7B, it uses less memory (2.5 GB vs. 2.8 GB) while also achieving a better PPL (6.55 vs. 6.96). For LLaMA-3-8B, the combined model is 41% better in PPL(11.72 vs 16.58) at a comparable memory size (4.5 GB vs. 4.6 GB).
>
> Summary: The value of Decomper is that it provides a new, composable tool for the compression ecosystem. By solving representation drift, it makes low-rank decomposition a viable technique that, when combined with quantization, achieves a superior compression-performance trade-off than was previously possible.

---

> > ### Comment · Reviewer_cKn2 · 2025-11-27
> > **Remaining Concerns on Practical Utility and Metric Reliability**
> >
> > Thank you for the detailed and very clear response.
> >
> > The authors have addressed most of my concerns. I still have some reservations regarding the practical usefulness of the method, but this does not affect the contribution or significance of the work within this research area. I will increase my score.
> >
> > My remaining concern is that, although the proposed method combined with 4-bit quantization achieves lower memory usage and only minor PPL degradation, perplexity is often not a convincing metric in practice. For example, on Llama-3-8B, Decomper reduces GSM8K accuracy from 0.55 to 0.09 (despite being stronger than similar approaches). When further combined with quantization, the performance may drop even more. The fact that this method can be combined with quantization does not necessarily make it a good idea. It would be helpful for the authors to compare pure quantization versus Decomper+quantization on GSM8K.
> >
> > While this performance drop may be an inherent limitation of this line of methods, the authors should more clearly articulate the method’s limitations and intended application scenarios.

---

> ### Author Response · Authors · 2025-11-28
> **Response to Follow-up: GSM8K with Quantization and Application Scenarios**
>
> We sincerely thank the reviewer for the positive feedback and for raising the score. We greatly appreciate the insightful discussion.
>
> 1. GSM8K Performance: Quantization Analysis
>
> As suggested, we have conducted experiments comparing pure quantization versus Decomper+quantization on GSM8K. We include GPTQ-4bit/3bit/2bit to show the full spectrum of memory-performance trade-offs.
>
> | Model | **Original** | **Unquantized (BF16)** | | **Pure Quant.** | | | **Combined** | |
> | :--- | :---: | :---: | :---: | :---: | :---: | :---: | :---: | :---: |
> | | | SVD-LLM | **Ours** | GPTQ-4bit | **GPTQ-3bit** | GPTQ-2bit | SVD-LLM+4bit | **Ours+4bit** |
> | **LLaMA-3-8B** | | | | | | | | |
> | *Memory Size* | 15 GB | 12 GB | 12GB | 5.4 GB | **4.6 GB** | 3.8 GB | 4.5 GB | **4.5 GB** |
> | *GSM8K Acc.* | 0.55 | 0.03 | **0.09** | 0.22 | **0.02** | 0.00 | 0.02 | **0.05** |
> | **LLaMA-2-7B** | | | | | | | | |
> | *Memory Size* | 12.6 GB | 10 GB | 10 GB | 3.7 GB | **2.8 GB** | 1.9 GB | 2.5 GB | **2.5 GB** |
> | *GSM8K Acc.* | 0.15 | 0.02 | **0.02** | 0.15 | **0.05** | 0.00 | 0.02 | **0.02** |
>
> Observations:
> - Sensitivity: For LLaMA-3-8B, combining quantization does lead to a further drop (0.09 $\to$ 0.05). This aligns with recent findings that LLaMA-3 is particularly sensitive to low-bit quantization compared to LLaMA-2.
> - The Trade-off: The reviewer is correct that pure 4-bit quantization (0.22) preserves more capability than decomposition. However, once compression becomes aggressive (e.g., GPTQ-3bit), performance collapses to near-zero (0.02), similar to decomposition methods.
> - Complex procedural reasoning is the first capability to degrade under aggressive compression. Nevertheless, even the quantized Decomper (0.05) outperforms the unquantized SVD-LLM baseline (0.03) and 2-bit quantization (0.00). This confirms our method provides a more robust foundation than comparable structural baselines.
>
> 2. Articulating Limitations and Application Scenarios
>
> We fully agree with the reviewer's suggestion to clarify the intended use cases. We have revised the "Limitations and Future Work” Section to explicitly categorize scenarios:
> - Recommended Scenarios: Decomper is highly effective for latency-sensitive applications requiring general language understanding, knowledge retrieval, and open-ended generation. In these areas, it provides structural speedups with minimal quality loss.
> - Cautionary Scenarios: For brittle, high-precision tasks like complex mathematical reasoning (GSM8K), we explicitly advise that compression (both decomposition and aggressive quantization) should be applied with caution or paired with instruction tuning, as capabilities in these domains are extremely sensitive.
>
> We believe this specific guidance adds practical value for practitioners.

---

### Official Review · Reviewer_irUD · 2025-10-28

**Soundness:** 3
**Presentation:** 3
**Contribution:** 3
**Rating:** 6
**Confidence:** 4

**Summary:**

This paper identifies “representation drift” — the propagation and amplification of approximation errors in low-rank decomposition of LLMs — as a key cause of performance degradation. The authors propose Decomper, a zero-overhead compensation method that learns bias corrections to align the outputs of decomposed layers with the original model. Experiments across multiple LLMs and benchmarks show the method effectively mitigates performance drops while adding no inference cost.

**Strengths:**

S1: The paper is well-structured with logical flow, facilitating reader comprehension. Section 3 presents a step-by-step, tightly connected process from error quantification to propagation analysis, enhancing readability and reinforcing technical rigor.
S2: The diagnosis of “representation drift” combines theoretical analysis with empirical validation. The proposed Decomper mechanism features zero deployment overhead and demonstrates strong generalizability across SVD/PCA-based decomposition, quantization, and vision-language models.
S3: Experiments cover diverse models and benchmarks, with in-depth analysis validating robustness and efficiency compared to fine-tuning/matrix updates, ensuring thorough method evaluation.

**Weaknesses:**

W1: Novelty: Decomposition-based compensation has been extensively explored in this field, as evidenced by prior works such as FLAP and AFM cited in the paper. This study primarily tests similar approaches on low-rank decompositions without substantial conceptual advancement.
W2: Although the paper conducts numerous comparative experiments, it fails to adequately contrast its approach with several outstanding related studies it cites.
W3: Section 3 presents critical formulas but omits intermediate derivation steps, and the Appendix does not supplement them. The paper directly states the linear layer reconstruction loss expectation without showing how to expand the squared norm expectation into mean and covariance terms.
W4: Equation 3 contains critical issues: the left-hand side is defined as the drift of the L-th block, but the right-hand side sums drift from l=1 to L, creating a logical contradiction. The formula also fails to retain the expectation term or connect to the squared L2 norm, breaking the link between theoretical setup and propagation analysis.
W5: The paper compares Decomper with recovery fine-tuning and least-squares matrix updates but omits critical experimental details. It remains unclear which dataset (WikiText-2 or C4) was used for fine-tuning/calibration, which benchmark the “Avg. Acc.” corresponds to, and how Decomper performs in domain-specific scenarios where fine-tuning typically excels.

**Questions:**

Please refer to Weaknesses.

**Details Of Ethics Concerns:**

N.A.

---

> ### Author Response · Authors · 2025-11-19
> **Part 1: Conceptual Novelty and Theoretical Clarifications (Q1-Q4)**
>
> We sincerely thank the reviewer for your positive assessment and constructive feedback. We are very encouraged that the reviewer found the paper "well-structured with logical flow," appreciated the "step-by-step, tightly connected process" of our analysis, and recognized the "strong generalizability" and "thorough method evaluation."
> We appreciate the detailed critique, which has identified several key areas for clarification and correction. We address each question below.
>
> > Q1&Q2: The compensation idea is common in this field. Inadequately Comparison with Related Works.
>
> A1&2: We acknowledge the need to better distinguish our work. Decomper is fundamentally distinct from prior art in three key aspects:
> 1. Problem Diagnosis: We are the first to identify, formalize, and empirically demonstrate "representation drift" as the specific root cause of failure for low-rank decomposition. This is a different error mode than that addressed by pruning (FLAP) or feature-space decomposition (AFM).
> 2. Methodology (vs. FLAP): FLAP uses a static heuristic ($c \approx \mathbb{E}[W_{\text{pruned}}x]$) to correct the mean shift. As analyzed in Sec 3.2, this fails for decomposition because it ignores the dominant variance error ($\operatorname{Tr}(E \Sigma E^\top)$) and non-linear distortions. In contrast, Decomper uses a learned optimization (Eq. 6) that empirically corrects for all error sources. Our experiments show Decomper consistently outperforms FLAP (e.g., LLaMA-2-7B PPL 8.07 vs. 9.21), validating that the heuristic is insufficient.
> 3. Orthogonality (vs. AFM): AFM is a feature-space decomposition method. Decomper is a decomposition-agnostic compensation framework. Appendix Table 11 shows that applying Decomper on top of AFM yields significant further gains (PPL 8.41 $\to$ 6.70), proving our method is a complementary tool, not a competing alternative.
>
> We are revising the Introduction and Related Work sections to explicitly articulate these fundamental conceptual and methodological distinctions.
>
>
> > Q3: Omitted Derivations in Section 3.
>
> A3: We apologize for omitting this standard derivation for brevity. Let $W_r$ be its rank-$r$ truncated approximation and $E = W - W_r$ the residual.
> The derivation from $\mathbb{E}[||E x||_2^2]$ to $||E\mu||_2^2 + \operatorname{Tr}(E \Sigma E^\top)$ relies on the definition of covariance, $\mathbb{E}[xx^\top] = \Sigma + \mu\mu^\top$, and trace properties.
>
> Full Derivation:
>
> We begin with the expectation of the squared norm:
>
> $$
> \mathcal{L}(W_r) = \mathbb{E}[||Wx-W_r x||_2^2] = \mathbb{E}[||E x||_2^2]
> $$
>
> **Step 1:** Express norm as trace
>
> $$
> = \mathbb{E}[\operatorname{Tr}((E x)(E x)^\top)] \quad (\text{since } ||v||_2^2 = \operatorname{Tr}(vv^\top))
> $$
>
> **Step 2:** Expand and apply linearity
>
> $$
> = \mathbb{E}[\operatorname{Tr}(E x x^\top E^\top)] = \operatorname{Tr}(\mathbb{E}[E x x^\top E^\top])
> $$
>
> **Step 3:** Use covariance definition
>
> $$
> = \operatorname{Tr}(E \cdot \mathbb{E}[x x^\top] \cdot E^\top) = \operatorname{Tr}(E (\Sigma + \mu\mu^\top) E^\top)
> $$
>
> **Step 4:** Separate terms and simplify
>
> $$
> = \operatorname{Tr}(E \Sigma E^\top) + \operatorname{Tr}(E \mu\mu^\top E^\top)
> $$
>
> **Step 5:** Final simplification
>
> $$
> = \operatorname{Tr}(E \Sigma E^\top) + \operatorname{Tr}((E\mu)(E\mu)^\top) = \operatorname{Tr}(E \Sigma E^\top) + ||E\mu||_2^2
> $$
>
> Thus we obtain the desired result: $\mathbb{E}[||E x||_2^2] = ||E\mu||_2^2 + \operatorname{Tr}(E \Sigma E^\top)$.
>
> We will include this full derivation in the Appendix.
>
> > Q4: Issues in Equation (3).
>
> A4: The reviewer is correct. We sincerely apologize for this critical typo and the resulting confusion. Our intended purpose was to conceptually illustrate how approximation errors propagate and amplify through the network's layers.
>
> The correct formulation we now use is:
>
> $$
> \mathcal{D}_{drift}^{L} = \mathbb{E}[||H^L - H_r^L||_2^2]
> $$
>
> $$
> \approx \mathbb{E}\left[|| \sum_{l=1}^{L} (PropagatedError_l) \cdot (LocalError_l) ||_2^2\right]
> $$
>
> Where:
> - $LocalError_l = \sum_{i \in \mathcal{W}} \frac{\partial \mathcal{F}^l}{\partial W^{i}} E^{i}$ represents the error introduced at layer $l$
> - $PropagatedError_l = \prod_{j=l+1}^{L} \frac{\partial H^j}{\partial H^{j-1}}$ captures how errors amplify through subsequent layers
>
> This motivates our solution: we must suppress the "Local Error" at its source in every layer and break the propagation chain of the "Propagated Error".

---

> ### Author Response · Authors · 2025-11-19
> **Part 2: Clarifications on Experimental Settings (Q5)**
>
> > Q5: Missing Experimental Setting Details.
>
> A5: We apologize for not making these details prominent. We clarify them here and in the revised paper.
> - Dataset: As stated in Section 4.3, Paragraph 1, all methods (FT, LS, and our Decomper) were calibrated using a general-domain corpus WikiText to ensure a fair comparison of the methods' recovery capabilities, not their ability to leverage specific data.
> - "Avg. Acc.": This refers to the average accuracy across the commonsense reasoning and knowledge intensive benchmarks used throughout the paper (BoolQ, PIQA, WinoG, HellaS, ARC-e, ARC-c, OBQA, and MMLU). This is the same metric used in Table 1.
> - Domain-Specific Scenarios: This is a fair point. Our study focuses on general-purpose compression and recovery. We hypothesize that if the goal was domain adaptation, FT with in-domain data would likely excel. However, our experiment (Table 5) shows that for general recovery on a general-domain corpus, our lightweight compensation is more effective.

---

### Official Review · Reviewer_9JKN · 2025-10-29

**Soundness:** 3
**Presentation:** 2
**Contribution:** 3
**Rating:** 4
**Confidence:** 4

**Summary:**

This paper studies the low-rank decomposition of LLMs. The authors discuss the layerwise decomposition and show that the error from layerwise decomposition can accumulate over different layers. They propose to add a bias term to compensate for the additional error. The resulting optimization is solved using gradient descent. The authors also provide numerical experiments over a wide range of baselines and benchmark datasets.

**Strengths:**

- The paper studies an important problem
- The numerical experiments are extensive

**Weaknesses:**

I think the writing can be improved. Some examples:

- Proposition 1 is quite handwavy. What is random? What is variance? With respect to what distribution? The results should be written properly.

- Why should one care about (3)? We only care about the final error, not the average layerwise error.

- It is not immediately clear how problem (6) is solved. One has to look into the appendix to find the algorithm, which I don't think it is referred to in the main text.

- Table 4 is missing dense baselines. I'm not sure how accurate the comparisons in the section named "Contribution of Compensation Strategy" are.

**Questions:**

Some of the models used in the experiments are rather old (OPT, Llama 2). Can the authors please present more benchmarks for newer models?

---

> ### Author Response · Authors · 2025-11-19
> **Part 1: Clarifications on Theory, Presentation, and Reference Models (Q1-Q4)**
>
> We thank the reviewer for your time and valuable feedback. We are encouraged that the reviewer finds our work to be on an "important problem" and that our "numerical experiments are extensive."
> The reviewer's main concerns appear to be related to the clarity of our writing and presentation, which we agree can be improved. We believe the issues raised are addressable matters of clarification rather than fundamental flaws. We will address each point below and commit to revising the paper to improve its clarity.
>
> > Q1: Clarity and proper writing about Proposition 1.
>
> A1: We apologize for the lack of formalization in Proposition 1. This was an oversight in our attempt to be concise. We will formally clarify this in the revision.
> - The "Randomness" refers to the sampling of inputs $x$, where $x \sim \mathcal{D}$ and $\mathcal{D}$ is the distribution of input activations.
> - "With respect to what distribution?": The expectations $\mathbb{E}[\cdot]$ and the statistics (mean $\mu$, covariance $\Sigma$) are all taken with respect to this input distribution $\mathcal{D}$.
> - "What is variance?": The "variance of the residual error" we refer to is specifically the term $\operatorname{Tr}(E \Sigma E^\top)$, which is the component of the uncompensated error (Eq. 1) that depends on the input covariance $\Sigma$.
> - Purpose: Proposition 1 illustrates that even with a theoretically optimal constant compensation $c^*$, the residual error $E_x[... ]$ remains bounded by the variance of the original error $\mu$. This provides theoretical support for our subsequent argument on "why merely matching the mean is insufficient."
>
> > Q2: Relevance of Equation (3).
>
> A2: We appreciate this comment.
> The purpose of Equation 3 is as a diagnostic tool and a source of motivation.
> - Diagnosis: It provides a first-order approximation of how the final error is constructed, demonstrating that the total error is not a simple summation of average or local errors.
> - Motivation: It illustrates that the errors from early layers (small $l$) are amplified the most by the product of Jacobians ($\prod \frac{\partial H^j}{\partial H^{j-1}}$). This analysis directly motivates our core idea: we must suppress the "Local Error" at its source (in every layer) to prevent it from being introduced and subsequently amplified.
> We will make it clear that Eq. 3 is an analysis that motivates our method, not the method or objective itself.
>
> > Q3. Solution of Equation (6) is not immediately clear.
>
> A3: We apologize for this poor signposting. The reviewer is correct that the details are in Algorithm 1 in the Appendix, but we failed to reference it properly in the main text.
> Problem (6) is solved by:
> - Freezing all original model weights ($W, W_r$).
> - Initializing the compensation vector $c$ (e.g., as zeros).
> - Iterating through the small calibration dataset and optimizing only $c$ using gradient descent (specifically, AdamW, as stated in Sec 4, Para 1) to minimize the block-level L2 loss.
> - This is done sequentially, block by block, from the first layer to the last.
>
> We will add a paragraph to Section 3.2 that explicitly describes this process and adds a clear reference to Algorithm 1 in the Appendix.
>
> > Q4. Missing dense baselines in Table 4.
>
> A4: This is a good suggestion that provides valuable context. Table 4 is intended to isolate the contribution of our compensation mechanism by comparing w/ Compensation directly against w/o Compensation. These dense model numbers are available in our other tables (e.g., Tables 1, 2, 3). We will add a "Dense Model" reference column to Table 4 in the revised version.

---

> ### Author Response · Authors · 2025-11-19
> **Part 2: Justification for Model Selection and Generality Analysis (Q5)**
>
> > Q5: Old Models like OPT and LLaMA-2 in the Experiments.
>
> A5: We appreciate the suggestion to use the latest models. We would like to gently clarify two key points about our model selection.
> 1. OPT and LLaMA-2 Were Chosen Deliberately for Rigor and Generality.
> Our inclusion of OPT and LLaMA-2 was a deliberate choice for two essential reasons:
>
>     (1) Architectural Diversity: This selection was also crucial for testing the generality of our method. LLM architectures are not monolithic. By including these models, we tested our compensation mechanism against critical architectural variations:
>     - Attention Mechanism: LLaMA-2 uses the Multi-Head Attention (MHA), whereas LLaMA-3 and QWen use the more recent Group Query Attention (GQA).
>     - FFN Structure: The OPT series uses a different FFN architecture compared to the SwiGLU/SiLU variants found in the LLaMA and QWen families.
>
>     (2) Fair Benchmarking: These models are foundational benchmarks used extensively in the LLM compression literature (e.g., by SVD-LLM, FLAP, SliceGPT, etc.). Using them allows for a direct and fair comparison with prior art, which is essential for placing our contribution in context.
>
> 2. We include 2024 and 2025 models:
>
>     (1) LLaMA-3-8B (released April 2024): This model is featured prominently in Tables 1, 3, and 4.
>
>     (2) QWen2.5-VL (released January 2025): We benchmark this recent vision-language model in Appendix Table 12, demonstrating our method's versatility across modalities.
>
> We hope we can provide a comprehensive and robust validation of our method's generality between different model architecture (MHA vs. GQA, different FFN), modalities, and release dates.

---

> > ### Comment · Reviewer_9JKN · 2025-11-25
> >
> > Thank you for your response.
> >
> > Re Proposition 1, I'm somehow even more confused. Please correct me if I'm wrong, but something does not add up. If you choose $c$ to minimize $||Wx - W_rx -c ||^2$, then the error is always zero:
> >
> > $E_x \min_c ||Wx - W_rx -c ||^2 = 0$
> >
> > If you mean out of sample error:
> >
> > $E_{x,y}  ||Wy - W_ry -c^* ||^2, c^* = Wx - W_rx$
> >
> > then you have to calculate
> >
> > $E_{x,y}  ||(W-W_r)(y-x) ||^2$
> >
> > which I believe you are claiming is equal to
> >
> > $E_{y}  ||(W-W_r)(y-E_x(x)) ||^2 = E_{y}  ||(W-W_r)(y-\mu) ||^2.$
> >
> > This does not seem correct to me. Could you please clarify? Given this is a mathematical proposition, if this issue is not resolved I have to reduce my score.
> >
> > Re experiments: I maintain that state of the art models are missing (DeepSeek ones, Qwen3, Kimi, Mistral). In addition, though not an issue with me, neither GSM8K is an advanced reasoning benchmark (you should consider AIME for example), nor Llama 2/3 are good reasoning models (you should consider Qwen3 or DeepSeek models).

---

> ### Author Response · Authors · 2025-11-26
> **Response to Follow-up: Clarification on Prop 1 & New Mistral-7B Results**
>
> We sincerely thank the reviewer for the continued engagement.
>
> Re Proposition 1: We clarify that the zero-error scenario mentioned by the reviewer is possible if the input distribution collapses to a single point (i.e., zero variance). Since real-world inputs inherently vary, a bias can only correct the mean shift, leaving the variance-dependent error unaddressed. This limitation directly motivates our proposed method.
>
> Re Experiments: We fully accept the suggestion to include state-of-the-art models. We have prioritized and completed experiments on Mistral-7B. The results of 20% compression ratio (see table below) confirm that our Decomper consistently outperforms baselines on modern architectures.
>
> | Model     | ppl | average | boolq | piqa | winogrande | hellaswag | arc_easy | arc_challenge | openbookqa | mmlu |
> |-----------|-----|---------|-------|------|------------|-----------|----------|---------------|------------|------|
> | Original Model   | 5.25 | 69.79   | 82.91 | 82.1 | 74.05      | 81.52     | 78.95    | 52.92         | 44         | 61.9 |
> | LLMPrun.  | 8.77 | 55.31   | 63.12 | 75.5 | 68.01      | 66.53     | 60.05    | 34.74         | 40         | 34.5 |
> | SliceGPT  | 8.22 | 42.68   | 38.63 | 59.96 | 58.93     | 45.95     | 47.85    | 30.05         | 32         | 28.1 |
> | SVD-LLM   | 7.26 | 52.88   | 71.07 | 67.05 | 66.05      | 52.92     | 61.79    | 35.02         | 36         | 33.1 |
> | Ours      | **6.58** | **59.82**    | 70.19 | 76.63 | 68.93      | 66.35     | 72.34    | 41.23         | 41         | 41.9 |
>
>
> We are actively conducting experiments on the other suggested models and benchmarks and will update the results as soon as possible.

---

> > ### Author Response · Authors · 2025-11-30
> > **Response to Follow-up: Qwen3-8B Results and Significantly Harder Reasoning Benchmarks**
> >
> > 1. Qwen3-8B Results
> >
> > We have completed the experiments on Qwen3-8B as suggested. Our method demonstrates remarkable robustness on this latest architecture, effectively preventing performance collapse even at high compression rates.
> >
> > For example, at 30% ratio, our Decomper **recovers Perplexity to 14.85 (vs. 20.03)**, and prevents collapse, **boosting MMLU by 51% (28.6 $\to$ 43.4)** and **recovering average accuracy by ~20% (46.5 $\to$ 55.7)**.
> >
> > | Ratio | Method | PPL | Avg. | BoolQ | PIQA | WinoG | Hella | ARC-e | ARC-c | OBQA | MMLU |
> > | :--- | :--- | :--- | :--- | :--- | :--- | :--- | :--- | :--- | :--- | :--- | :--- |
> > | **20%** | SVD-LLM | 13.86 | 56.91 | 79.08 | 69.42 | 61.09 | 54.93 | 57.83 | 39.59 | 36.4 | 42.7 |
> > | | **Ours** | **13.33** | **58.99** | 77.89 | 69.59 | 64.01 | 57.88 | 66.84 | 41.72 | 35.0 | **46.6** |
> > | **30%** | SVD-LLM | 20.03 | 46.45 | 63.12 | 62.30 | 54.62 | 41.46 | 45.71 | 28.16 | 29.8 | 28.6 |
> > | | **Ours** | **14.85** | **55.66** | 75.05 | 67.14 | 61.96 | 52.98 | 61.28 | 37.03 | 34.2 | **43.4** |
> > | **40%** | SVD-LLM | 42.82 | 38.25 | 37.86 | 56.75 | 52.33 | 34.13 | 34.22 | 25.09 | 27.4 | 25.1 |
> > | | **Ours** | **19.06** | **48.51** | 62.91 | 61.37 | 58.88 | 45.06 | 51.43 | 30.12 | 29.8 | **32.1** |
> >
> > These results on suggested SOTA models (Mistral & Qwen3) strongly validate the generalizability and effectiveness of our approach.
> >
> > 2. Regarding Elite Mathematical Reasoning Benchmarks (e.g., AIME)
> >
> > While we recognize AIME as a premier benchmark, it represents an **Olympiad-level** challenge that is significantly harder than standard benchmarks. Our analysis on GSM8K already indicates that procedural reasoning is inherently brittle under any compression (whether quantization, pruning, or decomposition). We expect zero-shot compression without instruction-tuning would yield **negligible scores across all methods**.
> >
> > Therefore, we believe the current priority is to **stabilize fundamental capabilities** (as reflected in our strong PPL, Common Sense, and Knowledge recovery) before tackling these **hyper-fragile, elite reasoning frontiers**. This approach is consistent with the literature, where compression methods typically focus on general-purpose benchmarks. We have included a discussion on "Limitations and Future Work" section.

---

### Official Review · Reviewer_a5jo · 2025-11-05

**Soundness:** 3
**Presentation:** 3
**Contribution:** 3
**Rating:** 6
**Confidence:** 5

**Summary:**

This paper identifies "representation drift" as a key problem in low-rank decomposition of LLMs, where approximation errors accumulate and amplify non-linearly through deep transformer layers, progressively distorting internal representations. The authors propose "Decomper," a compensation mechanism that learns bias vectors for each decomposed linear layer to align decomposed block outputs with original counterparts. The method is optimized on a small calibration set and adds zero inference overhead by fusing learned compensation into existing bias terms. Extensive experiments on OPT, LLaMA-2, LLaMA-3, and QWen2.5-VL etc, are shown.

**Strengths:**

- The core insight about error propagation through Equation 3 is well-motivated and provides theoretical grounding for the empirical phenomenon
- Proposition 1 offers a clear bound on the expected local error after compensation
- Experimental validation is comprehensive across multiple model families, scales, and benchmarks and the ablation studies convincingly isolate the contribution of the compensation mechanism

**Weaknesses:**

- The core idea of learning bias corrections is incremental—FLAP (An et al., 2024) already uses bias correction for pruning, and this work essentially adapts it for decomposition
- No comparison with other alignment strategies (e.g., feature matching, KL divergence minimization between distributions)
- The theoretical analysis in Section 3.1 uses first-order Taylor expansions without discussing when this approximation is valid or quantifying higher-order terms
- Proposition 1's proof assumes convergence to c* = Eμ but doesn't address the non-convex optimization landscape mentioned in Section 3.2
- The claim that Equation 6 "consistently converges to a strong local optimum" lacks empirical evidence (e.g., convergence curves, sensitivity to initialization)
- The theoretical closed-form solution c* = Eμ is dismissed as insufficient, but no rigorous analysis explains why the learned bias systematically outperforms this baseline
- Missing analysis of how the compensation mechanism interacts with different compression ratios per layer (mentioned but not studied)


Minor:
- Figure 2's caption could better explain what "23rd Transformer block" represents (out of how many?)
- The connection between the local error term (Equation 1) and the block-level propagation (Equation 3) could be made more explicit
- Section 3.2 introduces multiple ideas (theoretical c*, averaging trick, learned bias) that feel somewhat disconnected
- The paper claims "zero-cost deployment" but doesn't discuss memory overhead of storing compensation vectors during training or calibration time costs
- Some experimental details are unclear: what is "data-whitening SVD" as distinct from vanilla SVD?

**Questions:**

- Neat but not very well explained:why does the learned bias outperform the closed-form solution?
- Which layers contribute most to drift? Are all layers equally important to compensate? How does compensation allocation across layers affect results?
- What is the wall-clock time and memory cost of the compensation optimization phase? How does this scale with model size?
- Figure 2 shows drift recovery, but can you quantify the alignment more rigorously (e.g., KL divergence, Wasserstein distance between original and compensated distributions)?
- Can you provide error bars or confidence intervals for the main results (Table 1-3) to assess statistical significance?

---

> ### Author Response · Authors · 2025-11-19
> **Responses to Questions**
>
> We sincerely thank the reviewer for the detailed and constructive comments. We address all questions and concerns below.
>
> > Q1&Major Concern 6: Why does the learned bias outperform the closed-form solution $c^ = E\mu$?
>
> A1: The "closed-form" solution $c = E\mu$ is a linear heuristic that only corrects the mean error $\mathbb{E}[||Wx - (W_r x + c)||_2^2]$. Our learned bias  $\hat{c}$ (Eq. 6) is superior because:
> - Variance Correction: It corrects for the input-dependent variance error ($\operatorname{Tr}(E \Sigma E^\top)$), which $E\mu$ ignores.
> - Non-linear Adaptation: It minimizes errors at the block output, correcting distortions from non-linearities (SiLU, Softmax), which $E\mu$ cannot possibly account for.
>
> > Q2 & Major Concern 7: Which layers contribute most to drift? Are all layers equally important to compensate? How does compensation allocation across layers affect results?
>
> A2:  Thank you for this insightful question.
> - Drift Visibility & Error Source: While drift is most visible in deeper layers (due to accumulation), it originates in all layers (Eq. 3).
> - Compensation Strategy: Compensation is essential for all layers. Shallow layer compensation suppresses error amplification; deep layer compensation corrects accumulated drift.
> - Adaptive Allocation: Our optimization (Eq. 6) is inherently adaptive. It automatically learns larger compensation vectors for more sensitive layers.
>
> > Q3: What is the wall-clock time and memory cost of the compensation optimization phase? How does this scale with model size?
>
> A3: We thank the reviewer for raising this important question. Here is a summary of the costs:
> | Model | Wall-clock Time | Compensation Vectors Memory |
> |:---|:---|:---|
> | OPT-6.7B | ~ 50 mins | 0 |
> | OPT-13B | ~ 80 mins | 0 |
> | OPT-30B | ~ 3 h | 0 |
> | LLaMA-2-7B | ~ 75 mins | 4.4 MB |
> | LLaMA-2-13B | ~ 2 h | 6.5 MB |
> - Time Cost: Our advantage lies in the offline cost of the compensation optimization (Eq. 6). This process is highly efficient, completing for LLaMA-2-13B in just 2 hours. We optimize only the small vectors $c$ (e.g., $\mathbb{R}^{4096}$) with the model weights frozen; and most of the time is spent on periodic evaluations and checkpointing.
> - Memory Cost: For bias-equipped models, such as OPT series, our method requires no additional memory. For bias-free architectures like LLaMA, compared with models of more than ten GB, the compensation vector of several MB is negligible (0.03% of the original model).
> - Scaling: The time cost scales roughly linearly with model size, as it primarily involves a forward pass (to get outputs) and optimizing low-dimensional vectors.
>
> > Q4: Figure 2 shows drift recovery, but can you quantify the alignment more rigorously (e.g., KL divergence, Wasserstein distance between original and compensated distributions)?
>
> A4: This is a great suggestion. Figure 2 visually illustrates the drift. We agree that a rigorous metric is necessary to quantify the extent of recovery.
> We used the same input to compute the Wasserstein distance ($\downarrow$) between the original and decomposed latent distributions as suggested.
> | Model | Decom. w/o Compen. | Decom. w/ Compen. | Improvement |
> |-------|-------------------:|------------------:|------------:|
> | LLaMA-3-8B | 0.1324 | 0.0614 | **53.6%** |
> | LLaMA-2-7B | 0.2865 | 0.1749 | **38.95%** |
> | LLaMA-2-13B | 0.1820 | 0.0946 | **48.05%** |
> - Substantial Reduction: Our method consistently reduces the distributional divergence by a massive margin, ranging from 39% to nearly 54%.
> - Direct Correlation: This quantitatively confirms that our method successfully pulls drifting representations back to the original manifold. This alignment directly explains the performance recovery observed in our main experiments.
>
> > Q5: Can you provide error bars or confidence intervals for the main results (Table 1-3) to assess statistical significance?
>
> A5: Yes, we can. While we followed the community standard of using a fixed seed for fair comparison, we have now conducted multi-seed experiments (3 runs) to assess the variance of our method.
> The results on LLaMA-3-8B (20% compression ratio) show remarkable stability:
>
> | Metric | Seed 1 | Seed 2 | Seed 3 | Mean ± Std |
> |--------|--------|--------|--------|------------|
> | PPL | 11.172 | 11.085 | 10.956 | 11.071 ± 0.109 |
> | Common-sense Reasoning Average Accuracy | 0.5757 | 0.5733 | 0.5717 | 0.5736 ± 0.002 |
> | Knowledge-intensive MMLU | 0.373 | 0.371 | 0.365 | 0.370 ± 0.004 |
>
> The standard deviations are negligible compared to the performance margins we achieved over baselines. For example, the accuracy variance is only $\pm 0.2$%, whereas our method typically outperforms baselines by $>5$%. We will include the 3 runs' results in the final version.

---

> ### Author Response · Authors · 2025-11-19
> **Responses to Major Points**
>
> > Major Concern 1: The core idea of bias corrections is similar to FLAP (An et al., 2024).
>
> A1: Our work (low-rank approximation) is fundamentally distinct from FLAP (completely remove weight channels) and pioneers to:
> - Problem: Our work is the first to **formalize and empirically demonstrate representation drift (Figure 2)** as an unexplored critical cause of degradation in low-rank decomposition.
> - Method: We **derive compensation vector $c$** from representation alignment loss. This is fundamentally different from FLAP's heuristic, which simply computes the average contribution of pruned weights ($c \approx \mathbb{E}[W_{pruned}x]$).
> - Deployment: Embed $c$ into bias terms is a neat hardware-aware technique for **zero-cost deployment**.
> - As discussed in Page 5 Paragraph 4, **we did test the FLAP-style "averaging trick", and found it inadequate for decomposition**. The reason, as analyzed in our paper, is that this heuristic only cancels the mean error term ($E_{\mu}$) but ignores the significant, input-dependent variance term $\operatorname{Tr}(E \Sigma E^\top)$ and the non-linear errors from SVD truncation. Our learned approach (Eq. 6) optimizes $c$ to account for all these effects by minimizing the final block output error, leading to far superior performance.
>
>
> > Major Concern 2: Comparison with other alignment strategies.
>
> A2: We appreciate this suggestion. We chose L2 not just for its simplicity, but for its numerical stability and mathematical suitability.
> - KL/JS Divergences: Unsuitable. The hidden state contains negative values, causing **NaN** values.
> - While Wasserstein distance provides meaningful results, its computational complexity makes it impractical as a training objective. In contrast, L2 loss offers the ideal balance.
> - Cosine Similarity: The results show that **while Cosine Similarity is a reasonable alternative, our original L2 loss consistently outperforms it on both perplexity and accuracy**.
> | Model | Loss Function | 20% Compression | 30% Compression |
> |:---|:---|:---|:---|
> | LLaMA-2-7B | L2 Loss | PPL: 6.764; AVG. Acc: 59.43 | PPL: 8.074; AVG. Acc: 55.16 |
> | LLaMA-2-7B | Cosine Similarity Loss | PPL: 6.935; AVG. Acc: 59.09 | PPL: 8.318; AVG. Acc: 55.09 |
>
> We believe the key contribution is identifying the problem (drift) and the mechanism (bias $c$), while the choice of the loss function (L2 vs. Wasserstein distance/Cosine similarity) is a secondary, though interesting.
>
> > Major Concern 3: Discussing the validity of first-order approximation or quantifying higher-order terms.
>
> A3: We thank the reviewer for this sharp observation.
>
> We consider the **validity of first-order approximation**. We use the first-order Taylor expansion (Eq. 3) as a diagnostic tool to motivate the "suppress-at-source" intuition, not as a high-precision model. First-order analysis is a standard tool for interpreting deep networks (e.g., in model editing, quantization and pruning [1-4]). Our strong empirical results validate that this approximation successfully captures the dominant error propagation mechanism.
>
> [1] Kevin Meng, et al. Locating and Editing Factual Associations in GPT. NeurIPS 2022.
>
> [2] Hanyu Peng, et al. Deep Network Quantization via Error Compensation. TNNLS 2021.
>
> [3] Pavlo Molchanov, et al. Pruning convolutional neural networks for resource efficient inference. ICLR 2017.
>
> [4] Yongqi An, et al. Fluctuation-based Adaptive Structured Pruning for Large Language Models. AAAI 2024.
>
> > Major Concern 4 & 5: Proposition 1's proof assumes convergence to $c^ = E\mu$ but doesn't address the non-convex optimization. “Converges to strong local optimum" lacks empirical evidence.
>
> A4&5: These are related points that highlight a need for clarification.
> - Non-Convexity: We acknowledge Eq. 6 is non-convex. However, empirical evidence shows we can consistently find a strong local optimum.
> - Evidence: We added convergence curves to the Appendix. On LLaMA-3-8B, the loss drops steeply from 12.40 to 7.36 in 4 batches and converges at ~2.99 (a 76% reduction), confirming well-behaved optimization.

---

> ### Author Response · Authors · 2025-11-19
> **Responses to Minor Points**
>
> > Responses to Minor Points
>
> - Figure 2 Caption: Thank you. LLaMA-3-8B has 32 Transformer blocks. We will clarify this in the caption (e.g., "23rd Transformer block (out of 32)").
>
> - Eq 1 -> Eq 3 Link: We will make this link more explicit. The error term in Eq. 1, $\mathbb{E}[||E x||_2^2]$, is the expected value of the "Local Error" term ($\sum \frac{\partial \mathcal{F}^l}{\partial W^{i}} E^{i}$) in Eq. 3. Eq 1 analyzes the source of the error, and Eq 3 analyzes how it propagates.
>
> - Sec 3.2 Disconnected: We will rewrite section 3.2. We first present the theoretical optimum for a simple linear layer ($c^*=E\mu$). We then show why its naive practical implementation (the "averaging trick" $c=E\hat{\mu}$) is insufficient for decomposition. This motivates our final, superior approach: learning $\hat{c}$ via block-level output alignment (Eq. 6). We will refine the text to make this narrative flow clearer."
>
> - Zero-Cost Deployment: When we say "zero-cost deployment," we mean zero overhead during inference (as described in Q3). There is indeed an offline and one-off time and memory cost during the calibration phase, which we will clarify in the paper.
>
> - Data-Whitening SVD: We apologize for this ambiguity. This is a standard technique where SVD is applied to whitened activations. That is, given activations $X$, we compute the covariance $\Sigma = \operatorname{Cov}(X)$ and apply SVD to $W\Sigma^{1/2}$ instead of just $W$. This weights the SVD to better preserve the activation statistics. We will add a brief definition.

---

> ### Comment · Reviewer_a5jo · 2025-11-21
> **Thanks for the detailed response + two more quick questions.**
>
> Thank you for the detailed and very clear response!
>
> I have two more follow-up questions:
> 1. How are averages computed in Table 1? In Table 1, for Llama2-13B for a ratio of 30%
>     - For most rows it appears to be the average of the columns. Ex: for Decomper, the average is (63.55+73.72+66.06+65.71+54.89+38.2+34.1)/7 $\approx$ 56.60 (which is close to what is reported)
>     - However, for FLAP, the average is (64.37+72.42+63.93+62.44+49.37+39.2+33.2)/7 $\approx$ 54.99 which is very different from the 49.05 reported.
> 2. In Table 1 (or in the extended Table 14 in the appendix), for Llama-3 8B and OPT models (13B & 30B) why is the FLAP baseline not reported? It would be useful the explain the reason for that also in a footnote in the paper for clarity.

---

> ### Author Response · Authors · 2025-11-23
> **Response to Further Questions regarding Experimental Details**
>
> We sincerely thank the reviewer for your continued engagement.
>
> > Q1: How are averages computed in Table 1?
>
> A1:We thank the reviewer for their meticulous check. We clarify the discrepancy as follows:
> - Typo Correction: We acknowledge a transcription error for the FLAP LLaMA-2-13B (30%) average. The correct value is 54.29. We have verified all other entries in the table to ensure accuracy.
> - Calculation Formula: To conserve horizontal space, we condensed ARC-Easy and ARC-Challenge into a single column ("ARC") displaying their mean. However, the "Average" column is computed over all 8 original tasks (BoolQ, PIQA, WinoGrande, HellaSwag, ARC-e, ARC-c, OBQA, MMLU), rather than the 7 columns shown. This accounts for the slight numerical deviation observed when averaging the visible columns. We have added a footnote in the revised paper to explicitly clarify this calculation method.
>
> > Q2: Why is the FLAP baseline not reported for Llama-3 8B and OPT models?
>
> A2: The omission is due to architectural incompatibilities with the current implementations of these baselines:
> - OPT Series: As noted in the Appendix, methods like FLAP, LLM-Pruner, and ASVD are not currently adapted for the OPT architecture.
> - LLaMA-3: FLAP does not support the Grouped Query Attention (GQA) architecture used in LLaMA-3.
>
> We have added explicit footnotes in the revised paper to clarify these exclusions.

---

### Author Response · Authors · 2025-11-19
**Global Response**

Dear Reviewers,

We sincerely thank all reviewers (with IDs: a5jo, 9JKN, irUD, cKn2) for your valuable time and insightful comments aimed at improving our manuscript. We are pleased that the reviewers recognized the core motivation of our work (R-a5jo), the importance of the problem (R-9JKN), the clear logical flow (R-irUD), and the solid theoretical foundation (R-cKn2).

The reviewers' feedback was helpful for us to identify three key areas for strengthening: (1) the argument regarding novelty (particularly the comparison with methods like FLAP); (2) the rigor of the theoretical presentation (especially concerning Eq. 3 and Proposition 1); and (3) the experimental gap in comparisons on more challenging benchmarks (e.g., GSM8K).

In response to this feedback, **we have uploaded a revised manuscript**. To facilitate your review, **all major changes are highlighted in blue**. Specifically, we have made the following revisions:
- [Differentiation Clarification] We have substantially revised Section 2 (Related Work) to clearly and technically differentiate Decomper—which learns to mitigate the complex, input-dependent variance and non-linear drift introduced by SVD—from heuristics like FLAP, which only address the static mean shift introduced by pruning.
- [Theoretical Rigor] We are: (a) corrected a significant typesetting error in Eq. 3 (thanks R-irUD) to accurately reflect the recursive propagation of error; (b) provided a detailed derivation of the expectation expansion for Eq. 1 in the appendix (addressing R-irUD); and (c) formalized the assumptions for Proposition 1.
- [Complex Tasks] To address the reasonable concerns raised by R-cKn2, we have added new experiments on GSM8K. The results (detailed in our response to R-cKn2) indicate that while absolute drops are a field-wide challenge, Decomper provides performance recovery and achieve relatively better results, validating our drift mitigation thesis even in brittle scenarios.
- [More Models] In response to Reviewer 9JKN, we have expanded our evaluation to include state-of-the-art models, specifically Mistral-7B and Qwen3-8B. The results confirm that Decomper consistently outperforms baselines on these modern architectures (e.g., achieving a 51% relative improvement on MMLU for Qwen3-8B)
- [Relationship with Quantization ] To validate the practical value of Decomper (R-cKn2 Q3), we have included a new analysis demonstrating that Decomper is orthogonal and synergistic with quantization. Specifically, Decomper provides hardware-agnostic parameter reduction, which is complementary to the numerical memory reduction achieved by quantization. This synergy enables a superior compression-performance trade-off when combined.

We believe these revisions will significantly strengthen our paper and address the reviewers' concerns.

Sincerely,

Authors of submission #23511

---

### Author Response · Authors · 2025-11-30
**Summary of Rebuttal Progress**

Dear Area Chair,

We sincerely thank you for your considerable time and effort in this re-evaluation process under these exceptional circumstances. To assist your evaluation, we briefly summarize the consensus reached and the critical score update.

Core Contribution: Our work identifies "representation drift" as the key issue in low-rank decomposition and proposes a zero-overhead compensation mechanism, Decomper, which consistently outperforms baselines across diverse architectures and modalities.

1.Key Rebuttal Improvements
- Differentiation & Theory (Addressing R-a5jo, R-irUD, R-9JKN): Explicitly distinguished our learned compensation from static heuristics (e.g., FLAP), corrected Eq. 3, and formalized Proposition 1.
- Practicality Verified (Addressing R-cKn2): Added GSM8K benchmarks (Decomper achieves 9% acc vs. SVD-LLM's 3%) and demonstrated Synergy with Quantization (Decomper + 4-bit > 3-bit alone).
- Generalization: Added new validations on Mistral-7B and Qwen3-8B, and quantified drift reduction via Wasserstein distance (up to 53.6% reduction).

2.Explicit Reviewer Recognition
- **Reviewer cKn2 raised score from 4 to 6 on Nov 27, 12:38,** before the leak, confirming resolution of core concerns.
- Reviewer a5jo's shift from fundamental concerns to specific clarifications, after praising our response, indicates primary issues were resolved.
- Reviewer 9JKN's questions on Proposition 1 and modern models were addressed through mathematical clarification and new Mistral-7B and Qwen3-8B experiments.

Best regards,

The Authors of Submission #23511

---

### Meta-Review · Area_Chair_c2ZB · 2026-01-03

**Summary:**

This paper identifies compounding approximation error (or as the authors term it, "representation drift") as a cause of performance degradation in low-rank decomposition-based approaches to model compression. They then propose a fix that uses the bias to correct this at each layer. Reviewers were concerned about the lack of novelty relative to similar bias-modification approaches for other types of compression such as pruning and quantization, the validity and relevance of the mathematical analysis, and the completeness of the experiments.

**Reviewer Concerns:**

The authors partially addressed the incrementality concerns by claiming that the "representation drift" concept was itself a novel contribution, even if their solution has similarity to past work that used biases to correct other types of compression. It is unclear if this satisfied any reviewers, and it is not obvious if the concept itself is surprising as a contribution; for example, it has been observed in well-known model compression papers such as Kuzmin et al. (2023, Figure 8). It is also unclear if the concerns about the mathematical analysis were addressed, despite some discussion. It seems that concerns about the experiments were partially addressed, although new concerns were raised about severe degradation on reasoning benchmarks relative to quantization benchmarks.

## References
Kuzmin, Nagel, van Baalen, Behboodi, Blankevoort. *Pruning vs Quantization: Which is Better?* NeurIPS 2023.

**Reviewer Scores:**

It is unclear if from Reviewer a5jo's comments if they would have raised their score from a 6.
It seems unlikely from Reviewer 9JKN's comments that they would have raised their score from a 4.
It seems unlikely from Reviewer irUD's comments that they would have raised their score from a 6.
The authors' and reviewer's comments reveal that Reviewer cKn2 increased their score from a 4 to a 6.

---

### Decision · Program_Chairs · 2026-01-26

Reject